# Msx1$^+$ stem cells recruited by bioactive tissue engineering graft for bone regeneration

Xianzhu Zhang[1,2,3,6], Wei Jiang [1,3,6], Chang Xie[1,2,3,6], Xinyu Wu[1,3], Qian Ren[4], Fei Wang[1,3], Xilin Shen [1,3], Yi Hong[1,3], Hongwei Wu[1,3], Youguo Liao[1,3], Yi Zhang[1,3], Renjie Liang[1,3], Wei Sun[1,3], Yuqing Gu[1,3], Tao Zhang[1,3], Yishan Chen[1,3], Wei Wei[1,3], Shufang Zhang[1,3], Weiguo Zou[4,5] & Hongwei Ouyang [1,2,3,5] ✉

Critical-sized bone defects often lead to non-union and full-thickness defects of the calvarium specifically still present reconstructive challenges. In this study, we show that neurotrophic supplements induce robust in vitro expansion of mesenchymal stromal cells, and in situ transplantation of neurotrophic supplements-incorporated 3D-printed hydrogel grafts promote full-thickness regeneration of critical-sized bone defects. Single-cell RNA sequencing analysis reveals that a unique atlas of in situ stem/progenitor cells is generated during the calvarial bone healing in vivo. Notably, we find a local expansion of resident Msx1+ skeletal stem cells after transplantation of the in situ cell culture system. Moreover, the enhanced calvarial bone regeneration is accompanied by an increased endochondral ossification that closely correlates to the Msx1+ skeletal stem cells. Our findings illustrate the time-saving and regenerative efficacy of in situ cell culture systems targeting major cell subpopulations in vivo for rapid bone tissue regeneration.

Bone injury is common in the clinic, and critical-sized bone defects cannot be self-healed. The "golden standard" of clinical treatments for such defects is still based on the transplantation of autologous/allogeneic or artificial bone grafts. However, there are disadvantages, such as donor limitation, immune rejection, infections and poor osseointegration[1], that eventually lead to secondary surgery due to heterogeneous scar regeneration, delayed healing or even nonunion[2].

Current approaches using the biomineralized hydrogels[3] or osteogenic preconditioning[4] can hardly achieve full-thickness reconstruction of the critical-sized calvarial defects, in which two layers of cortical bone plate and trabecular bone/bone marrow in between exist[5], usually leading to the formation of immature bone tissues. The limitations of current strategies for repairing large bone defects have

led to increasing interest in using stem cell-based bone graft transplantation. The majority of stem cell-based treatments for bone repair are still being performed as pilot studies that mainly focus on the use of in vitro expanded mesenchymal stem cells (MSCs) in plastic-adherent 2D culture or microcarrier-based 3D culture[6]. However, the in vitro engineered bone grafts incorporating stem cells have shown limited potential for further clinical applications, due to the inadequate in vitro expansion efficiency[7,8], long culture times and limited in vivo survival rates[9,10]. In addition, the in vitro expanded or preconditioned MSCs are highly heterogeneous, and the subsequent in vivo lineage commitment cannot be precisely predicted after transplantation, necessitating transplantation by trial and error[11,12], and the underlying repair mechanisms for bone repair remain elusive.

[1]Dr. Li Dak Sum & Yip Yio Chin Center for Stem Cells and Regenerative Medicine, and Department of Orthopedic Surgery of the Second Affiliated Hospital, Zhejiang University School of Medicine, Hangzhou, China. [2]Department of Sports Medicine, Zhejiang University School of Medicine, Hangzhou, China. [3]Zhejiang University-University of Edinburgh Institute, Zhejiang University School of Medicine, and Key Laboratory of Tissue Engineering and Regenerative Medicine of Zhejiang Province, Zhejiang University School of Medicine, Hangzhou, China. [4]State Key Laboratory of Cell Biology, CAS Center for Excellence in Molecular Cell Sciences, Shanghai Institute of Biochemistry and Cell Biology, Chinese Academy of Sciences, University of Chinese Academy of Sciences, Shanghai 200031, China. [5]China Orthopedic Regenerative Medicine Group (CORMed), Hangzhou, China. [6]These authors contributed equally: Xianzhu Zhang, Wei Jiang, Chang Xie. ✉e-mail: hwoy@zju.edu.cn

An alternative strategy is the in situ delivery of soluble factors by recruiting and empowering endogenous stem/progenitor cells for more efficient tissue repair. Indeed, harnessing endogenous stem cells by recombinant growth factors for tissue regeneration has been extensively studied, and it can avoid the complicated extraction, purification, amplification, and quality control process that required for exogenous stem cell transplantation[13,14]. Previous studies have shown that activated stem cells derived from dental pulp[15], synovium[16], tendons[17] and meniscus[18] by local delivery of exogenous growth factors can effectively repair the targeted tissues. Thus, resident stem cells showed strong tissue-specific differentiation and directed regeneration potential, and did not require excessive in vitro procedures to maintain cell phenotypes and behaviors. Similarly, upon bone injuries, various stem cells are available and critical for efficient bone repair. Well-known treatments for bone defects include the local delivery of recombinant human bone morphogenetic protein 2 (rhBMP-2) soaked in collagen sponge carriers[19,20] or loaded in microspheres[21] that only focus on promoting the osteogenic differentiation of endogenous stem/progenitor cells. However, the therapeutic effect of these methods is largely affected by the harmful ectopic bone formation, osteoclast activation and soft tissue inflammation[22], as well as the requirements for additional mechanical stimuli. While the in vivo stem/progenitor cells during the bone regeneration process are also highly heterogeneous, biomedical engineering strategies using soluble factors by targeting in situ expansion of specific stem cell subpopulations with osteogenic potential remain challenging.

Recent studies have shown a direct contribution of skeletal stem cells (SSCs) to bone development and injury repair[23]. SSCs resident in bone tissues could be stimulated to proliferate locally and give rise to bone, cartilage, and stroma after injuries. An increasing number of studies have also shown that LepR[+24], Nes[+25], and Mx1[+26] SSCs can migrate to the injured site and contribute to enhanced new bone formation. However, the osteogenic subpopulations of SSCs that increase in number are rapidly dividing and short-lived after bone fracture, and are significantly depleted during aging[27,28]. The limited numbers of endogenous skeletal stem cells are insufficient for complete bone regeneration, especially in mature or elderly individuals[27]. Hence, soluble factors that could locally expand large numbers of specific subpopulations of stem cells with osteogenic potential in vivo to regenerate bone tissues are still need to be investigated.

Inductive factors have shown spatial-temporal control of skeletal development through multiple signaling pathways[29,30]. Neurotrophins and neurotrophic tyrosine kinase receptor type A (NGF-TrkA) signaling directs innervations in bone tissues[31], which are also required for osteochondral progenitor expansion during bone development and stress fracture repair[32]. Interestingly, after the in vitro expansion in serum-free medium that containing neurotrophic supplements (NSs), which include N2, B27, epidermal growth factor (EGF), fibroblast growth factor (FGF)-basic, insulin-like growth factor-1 (IGF-1), platelet-derived growth factor (PDGF) and Oncostatin M (OSM), MSCs form a menssphere with higher self-renewing ability[33] and exhibit non-adherent spheroidization characteristics with improved microcirculation properties[34]. Meanwhile, MSCs showed a dynamic proportion of nestin[+] staining in various locations within the bone marrow and in their long-term cultures. After the menssphere culture with NSs-containing medium, nestin[+] MSCs were demonstrated to be more inclined to differentiate into bone and cartilage forming cells than unsorted MSCs[25]. These findings indicate that with the optimization of neurotrophic supplements (NSs) for culture medium, distinct stem cell subpopulations could be induced and preserved for enhanced osteogenic potential for skeletal tissue regeneration. More importantly, when adopting soluble factors-containing culture medium for engineering an in situ culture system for local expansion of resident stem cell subpopulations, the corresponding mechanisms during the bone regenerative process at single-cell resolution are still not fully understood.

Therefore, we hypothesize that in situ delivery of NSs through a 3D-printed bioactive hydrogel graft could induce bone regeneration by promoting local expansion of skeletal stem cell subpopulations in vivo. In this study, we report a 3D-printed hydrogel scaffold incorporated with a cocktail of optimized NSs for use as an in situ cell culture system induces robust expansion of mesenchymal stem cells (MSCs) in vitro and successfully achieves a full-thickness regeneration of critical-sized calvarial defects in vivo. Notably, by single-cell RNA analysis, we show that implantation of the culture system greatly promoted the in situ recruitment and expansion of a specific Msx1[+] skeletal stem cell (SSC) subset. Moreover, the enhanced bone regeneration was accompanied by endochondral ossification that was closely correlated with Msx1[+] SSCs. In summary, this study illustrates the time-saving and regenerative efficacy of in situ stem cell expansion by a tissue engineering scaffold containing soluble factors for more efficient bone regeneration.

## Results

### The 3D printed hydrogel incorporating NSs induces robust in vitro expansion of MSCs with enhanced osteogenic potential

To investigate the role of soluble factors in promoting the expansion of mesenchymal stem cells, we performed cell culture experiments using growth medium with optimized neurotrophic supplements (NSs), which is a specialized cocktail of neurotrophic factors adapted from the culture of bone marrow mensspheres[25] and neural stem cells[35]. In previous studies used serum-free medium, we optimized the NSs-supplemented medium by depleting some unnecessary recombinant growth factors that could induce cell differentiation, and then combined with fetal bovine serum (FBS) for our final NSs formula for MSC expansion. When observed after 7 days of culture, human MSCs appeared to have a more elongated cell morphology and smaller cell attachment area (Supplementary Fig. 1a), as well as a significantly lower cell population doubling time (PDT) in culture with NSs-containing medium compared to a regular growth medium (Supplementary Fig. 1b, c). To further explore whether the higher cell proliferation with a stable MSC phenotype was achieved after NSs treatment, we tested the expression of the cell proliferation marker Ki67 and the cell senescence marker γh2AX (indicating DNA damage). Immunostaining showed that the expression of Ki67 was significantly increased while the expression of γh2AX was significantly reduced after culture with NSs when compared with the non-NSs control medium (Supplementary Fig. 1d–f). In response to these clear changes, we performed detection of cell surface markers on the expanded MSCs by cell flow cytometry. The results showed that cell products under both culture conditions were positive for CD105, CD29, and CD90 and negative for CD45 (Supplementary Fig. 1g), a minimal criterion for MSCs[36]. Moreover, compared to the non-NSs-expanded MSCs, the NSs-expanded MSCs were significantly more clonogenic (Supplementary Fig. 1h) and showed enhanced potential in chondrogenic and osteogenic lineages with a similar potential for adipogenesis (Supplementary Fig. 1i, j). The above results indicate that although the cell proliferation, morphology and attachment of MSCs were significantly changed after NSs treatment, the stem cell phenotype remained unchanged. To more directly identify the osteogenic potential of the rapidly expanded MSCs after NSs culture, after 3 days of pretreatment with NSs medium, we replaced the medium for osteogenic induction rather than an additional cell passaging process (Supplementary Fig. 2a). In addition to a robust increase in cell growth as indicated by cell numbers after 3 days of pre-treatment, osteogenic differentiation was assessed by alkaline phosphatase (ALP) staining at Day 10 and Alizarin Red S staining at Day 17. The results showed that they were both significantly enhanced after osteogenic induction, respectively (Supplementary Fig. 2b–d).

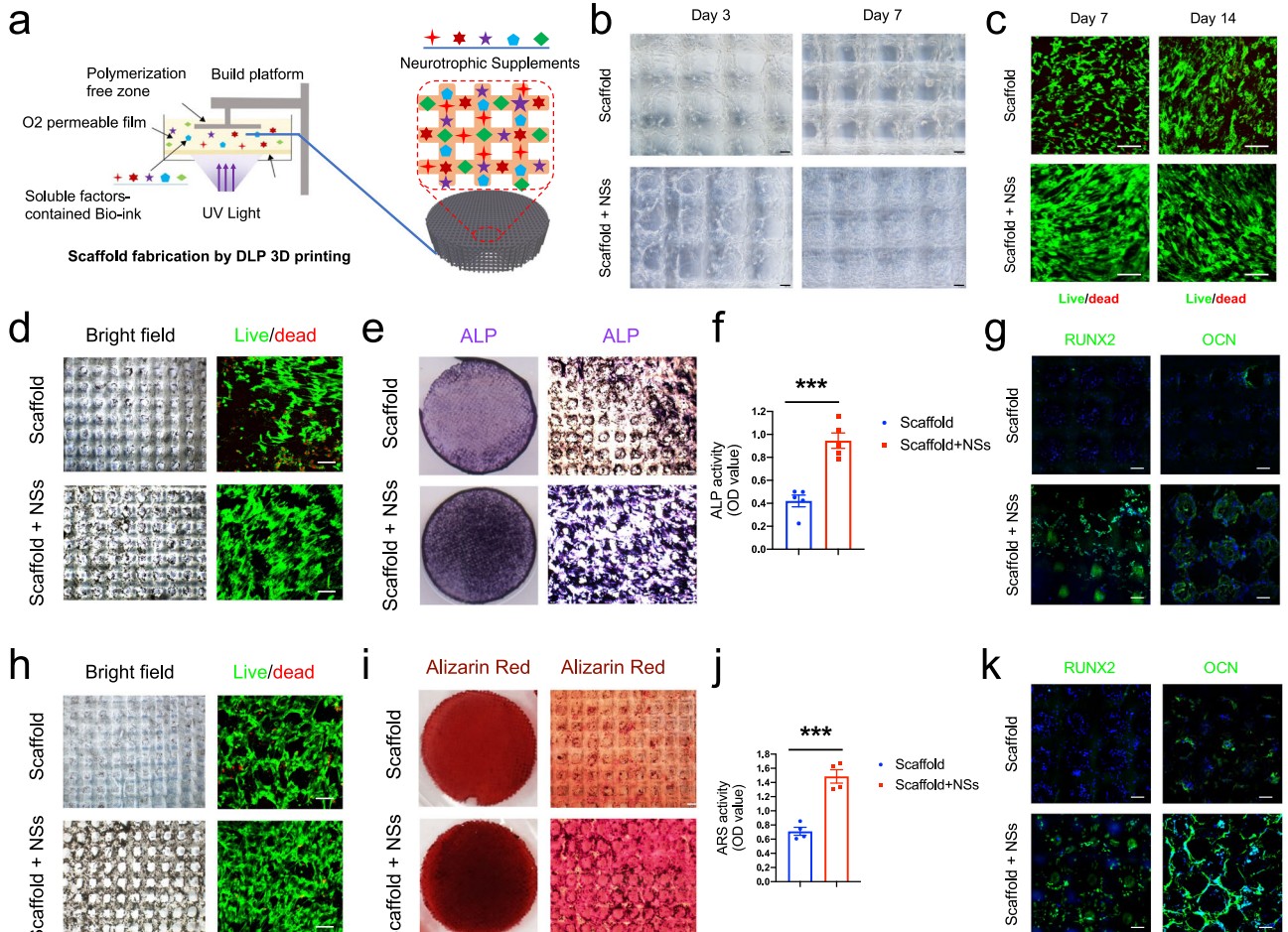

**Fig. 1 | 3D printed hydrogel scaffold with NSs enhances in vitro expansion and osteogenic potential of MSCs. a** Schematic diagram of NSs-loaded 3D printing gel scaffold by DLP technology. **b** Bright field image of MSCs distribution and cell morphology on the surface and inside of the gel scaffold and NSs-loaded gel scaffold for 3 days and 7 days, bar = 200 μm. **c** Live/Dead staining (Green, live cells; Red, dead cells) of MSCs expanded on gel scaffold and NSs-loaded gel scaffold for 7 days and 14 days, bar = 50 μm. **d** Bright field image and Live/Dead staining (bar = 200 μm, 50μm); **e** Gross view and **f** ALP staining, bar = 100μm, Exact *p* value calculated with unpaired *t*-test: ***p = 0.0003, *n* = 5 biologically independent samples; **g** immunostaining of RUNX2 and OCN after 7 days' osteogenic differentiation of expanded MSCs on gel scaffold after 7 days (bar = 30 μm). **h** Bright field image and Live/Dead staining (bar = 200 μm, 50 μm); **i** Gross view and **j** Alizarin Red S staining, bar = 100μm, Exact *p* value calculated with unpaired *t*-test: ***p = 0.0004, *n* = 4 biologically independent samples; **k** immunostaining of RUNX2 and OCN after 14 days' osteogenic differentiation of expanded MSCs on gel scaffold after 7 days (bar = 30 μm). Bone marrow stem cells are isolated from orthopedic individuals with written informed consent. All data represent mean ± SEM, and source data are provided as a Source Data file. At least three times of experiments were repeated independently.

It is known that the application of soluble factors requires in vivo delivery, and natural biomaterials provide with a better choice. To determine an appropriate carrier for further in situ NSs delivery, we applied a highly biocompatible, degradable and adhesive GelMA/HA-NB hydrogel, which was reported in our previous studies[37,38]. As GelMA/HA-NB has a photo-crosslinkable property, it has the advantage of being useable as a bio-ink in the photo-crosslink based bioprinting. Here in this study, by using digital light processing (DLP) 3D printing technology, we fabricated NSs-incorporated cylindrical hydrogel flakes with interconnected microchannels of 200 μm width and a layer thickness of 50 μm (cylindrical flake diameter: 8 mm; thickness: 2 mm) (Fig. 1a). Moreover, such 3D-printed hydrogel scaffolds provide with more inner space and surface area that are needed for sufficient cell migration and attachment. When loaded with NSs, the hydrogel scaffold can be used as an in vitro 3D gel culture system, and MSCs were seeded on the scaffold for in vitro expansion. The results showed that MSCs under different growth media migrated and distributed evenly along the inner surface of the backbone within the scaffold 3 days after seeding, while with NSs in the medium, MSCs were apparently expanded and densely attached across the pore structures 7 days after seeding (Fig. 1b). Live/dead staining further confirmed the increased cell number and high level of live cells cultured in medium with NSs during 2 weeks of expansion (Fig. 1c).

To test the osteogenic potential of the in vitro expanded MSCs by the gel culture system, we changed the growth medium to osteogenic medium after 7 days of expansion. Except for the increased cell number (Fig. 1d), we found that osteogenic differentiation was significantly upregulated in the NSs-loaded gel culture system, as indicated by both ALP staining (Fig. 1e, f) and immunostaining for Runx2 and OCN (Fig. 1g) for short-term (7 days) osteogenesis. Further extending the osteogenic induction time to 14 days, we found that the cells within the scaffold were more evenly distributed than those in the control group. There were also more abundant living cells on the outer and inner surfaces of the NSs scaffold, and the proportion of living cells was higher (Fig. 1h). In addition, NSs led to significantly higher levels of calcium nodules, as indicated by Alizarin Red S staining (Fig. 1i, j), and they promoted the expression of osteogenic markers of MSCs at the mature stage (Fig. 1k).

Together, these data suggested that 3D-printed scaffolds incorporating NSs successfully constructed an ideal in vitro cell culture gel

scaffold system for MSCs. When compared with the vehicle scaffold, the NSs-loaded cell culture gel scaffold can not only expand MSCs effectively and initiate osteogenic differentiation more quickly but also greatly enhance the osteogenic differentiation in the mature stage. These findings lay a theoretical foundation and provide an available approach for the in situ expansion and osteogenic induction of endogenous stem cells.

## The in situ culture system with NSs promotes full-thickness regeneration of critical-sized calvarial defects

To evaluate the potential of the local delivery of NSs for promoting bone regeneration, we first tested the in vivo cell compatibility and vascularization of the gel culture system. After subcutaneous implantation at the dorsal site, we found that the scaffold was gradually degraded within four weeks, and NSs incorporation accelerated this progression as shown by the obviously reduced gel area (Supplementary Fig. 3a, b). Different from the limited cells during in vitro experiments, the physiological microenvironment in vivo provides abundant cell types and numbers. Results showed that cells were evenly distributed inside the porous structure (Supplementary Fig. 3c), suggesting the availability of the gel scaffold for in vivo cell culture. As the bulk hydrogel naturally blocked the cell migration and became a thin layer before its complete degradation, multichannel structures harbored cells across the entire scaffold (Supplementary Fig. 3d). Notably, in the NSs-containing scaffold, we observed a large number of mixed cells, including the acute infiltration of monocyte-like cells, that migrated and expanded into the scaffold after 1 week of implantation. In the following histological assessment, we found that the scaffold was nearly replaced by fibroblasts and red blood cells, as well as endothelial cells (Supplementary Fig. 3d, e). In addition, we found immature collagen formed at week 2 (Supplementary Fig. 3f). Thus, the results demonstrated that the in situ culture system maintains high cell compatibility and promotes efficient in vivo vascularization.

We then transplanted the in situ culture system consisted of NSs-loaded 3D-printed gel scaffolds into a critical-sized calvarial defect of SD rats (Fig. 2a, b). 4 weeks after surgery, we found that the vehicle scaffold alone assists bone healing to a certain extent, while the NSs-loaded scaffold regenerated almost two-thirds of the entire defect area (Fig. 2c, e, f). Consistently, 12 weeks after surgery, the vehicle scaffold still had not fully promoted the healing of the defect, while the NSs-loaded scaffold basically completed the regeneration of the entire defect with a significantly higher bone mass, and good integration of newly formed (yellow area) and old (gray area) bones was observed (Fig. 2d, g, h). In addition to the increased bone mass, the histological structure and collagen component of new bone tissue were also evaluated. Remarkably, the results showed that full-thickness skull (internal and external layers of cortical bone; middle layer of trabecular bone) regeneration of the calvarial defect was achieved by the NSs-loaded scaffold (Supplementary Fig. 4a, b), other than a single layer of disordered immature bone in the vehicle scaffold or a thin layer of soft tissue in the control defect (Fig. 2i, k). The presence of the bone tissue newly formed by the NSs-loaded scaffold was further confirmed by Masson's Trichromic staining showing the coexistence of bone tissues with different maturities in the regenerated area (Fig. 2j, l). The results also showed that NSs promoted a mild increase in the expression of osteoblast marker Osteocalcin (OCN) (Fig. 2m, Supplementary Fig. 4c, d)and the neural growth factor receptor (NGFR) (Fig. 2m), as well as vascular endothelial cell marker CD31 (Fig. 2n) expression at 4 weeks after injury but significantly promote OCN (Fig. 2m) and CD31 (Fig. 2n) expression at 12 weeks after injury. The above results indicate that in situ delivery of NSs significantly restored the structural integrity and promoted functional regeneration of full-thickness calvarial bone tissue.

In conclusion, NSs can not only effectively promote the in vitro expansion of mesenchymal stem cells but also enhance the osteogenic potential of the expanded cell products. More importantly, the in situ delivery of NSs by a 3D-printed hydrogel scaffold promoted full-thickness regeneration at critical-sized calvarial defects, in addition to immature bone formation.

## The in situ culture system with NSs differentially regulates four bone repair-associated cell clusters

To determine differentially regulated cell subsets in response to the in situ culture system with NSs delivery, we performed single-cell RNA-sequencing analysis of the in situ cell clusters after implantation of the cell culture system for 1 week and 2 weeks (Supplementary Fig. 5a, b). Cells of the regenerative tissues were isolated, and mRNA libraries were prepared (10x Genomics) and sequenced (Fig. 3a). For quality control, cells with over 300 genes detected, 500 read counts and less than 20% of mitochondrial gene expression were kept. After filtering, we obtained data from 26016 cells: with 7749 cells in the defect group, and 7712 cells in the NSs-treated group of samples from 1 week after surgery; and 5420 cells in the defect group, and 5135 cells in the NSs-treated group of samples from 2 weeks after surgery. Following the Seurat pipeline[39], the data were log-normalized and then scaled. Uniform Manifold Approximation and Projection (UMAP) and t-distributed Stochastic Neighbor Embedding (t-SNE) were calculated to visualize cell heterogeneity in reduced dimensions.

Integrated analysis identified 33 subpopulations that occurred during the first two weeks of bone repair (Fig. 3b, c). All these subpopulations were clustered into four groups: an osteo-lineage cell population (expressing Col1a1 and Postn); an endothelial cell population (expressing Cdh5 and Vwf); a perivascular cell population (expressing Acta2 and Pdgfrb); and an immune-lineage cell population (expressing CD45 and CD14) (Fig. 3d–g). As the in situ culture system showed a strong advantage in promoting cell expansion of multiple cell types (Supplementary Fig. 5c, d), we next examined the differentially regulated cell subpopulations after the in situ delivery of NSs. Our results first revealed a more significant alteration in the proportion of osteogenesis-related stem/progenitor cells within the osteo-lineage cell population upon NSs delivery. However, the relative proportion of the stem/progenitor subclusters within both endothelial and perivascular cell population showed only mild differences in response to the NSs (Fig. 3h–j).

An additional significant response to NSs is an interesting shift in cluster contributions throughout the immune-lineage cell populations (Supplementary Fig. 6a–c). Specifically, the monocytes and CD8[+] T cells underwent a considerable decrease following 2 weeks of NSs delivery (Supplementary Fig. 6d). Since the immune-lineage cell population is beyond the focus of the present study, our subsequent data analysis mainly focused on the obviously regulated osteo-lineage cell subpopulations. Collectively, these data provide a single-cell resolution atlas of the regenerative-state of bone regeneration, which enables us to reveal cellular heterogeneity within various cell types, especially the osteo-lineage cells of interest.

## The in situ culture system with NSs regulates a distinct skeletal stem/progenitor cell atlas during bone regeneration

To further test whether the in situ culture system of a 3D-printed hydrogel scaffold with NSs could induce bone regeneration by modulating the skeletal stem/progenitor cell subpopulations in vivo, we first focused on the cellular composition and differential regulation of the osteo-lineage cell subpopulations caused by NSs. Within osteo-lineage cells, we distinguished eight major subsets by sub-clustering analysis. In particular, they are: (1) Cycling MSCs (expressing Mki67, Ube2c and Cspg4); (2) two SSC subsets (expressing Itga5, CD200, and Prrx1); (3) two Mesenchymal Progenitor Subsets (expressing Cxcl12, Angptl1, Pdgfra and Sfrp2, Osr2); (4) Fibroblast (expressing S100a4 and Fn1); (5) two Osteoblast subsets (Alpl, Runx2, and Ogn). (Fig. 4a–c) Among these osteo-lineage cell subpopulations, we noticed that there

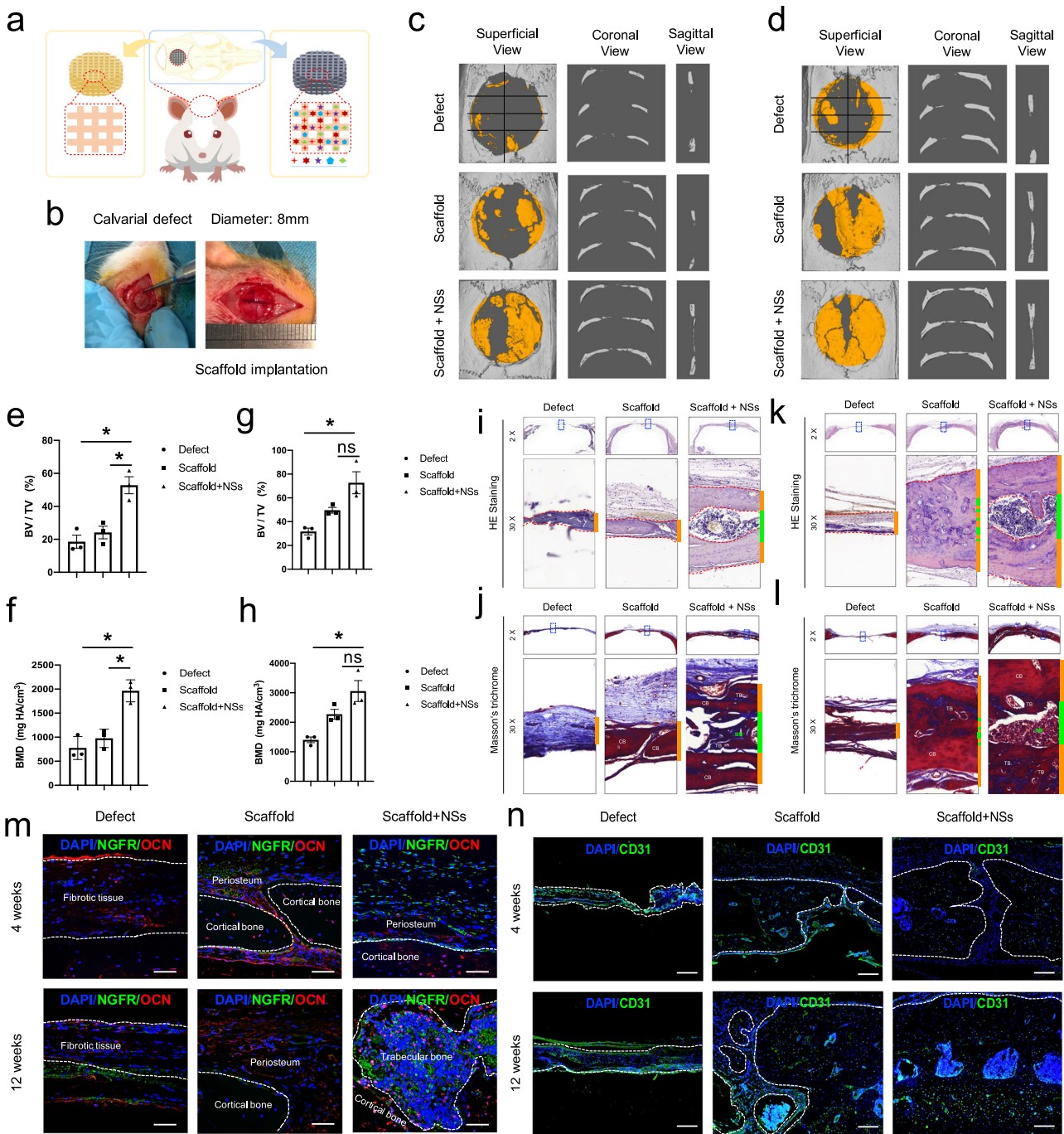

**Fig. 2 | Transplantation of the 3D printed hydrogel scaffold with NSs promotes rapid regeneration of critical-sized calvarial defect. a, b** Illustration of the implantation of NSs-loaded 3D printing gel scaffold before and after calvarial defect surgery. **c, d** μ-CT evaluation of bone regeneration in Defect control (Defect), 3D printed scaffold (Scaffold), 3D printed scaffold containing NSs (Scaffold + NSs) groups at 4 weeks and 3 12 weeks post-surgery of calvarial defect (the yellow area indicates new bone formation), respectively. $n = 3$ rats for per group and per time point. **e** Bone volume fraction (BV/TV) (Exact $p$ value calculated with one-way ANOVA Tukey's multiple comparisons test: *$p = 0.0218$, *$p = 0.0391$); and **f** Bone mass density (BMD) (Exact $p$ value calculated with one-way ANOVA Tukey's multiple comparisons test: *$p = 0.0113$, *$p = 0.0214$) of regenerated tissues in defect area at 4 weeks after calvarial defect. **g** Bone volume fraction (BV/TV) (Exact $p$ value calculated with one-way ANOVA Tukey's multiple comparisons test: **$p = 0.0076$, $p = 0.0532$); and **h** Bone mass density (BMD) (Exact $p$ value calculated with one-way ANOVA Tukey's multiple

comparisons test: **$p = 0.0045$, $p = 0.0582$) of regenerated tissues in defect area 12 weeks after calvarial defect; $n = 3$ biologically independent rats. **i, j** HE staining and Masson' Trichromic staining of paraffin sections in Defect, 3D Scaffold, Scaffold + NSs groups at 4 weeks after calvarial defect (bar = 500 μm at low magnification and bar = 50 μm at high magnification). **k, l** HE staining and Masson' Trichromic staining of paraffin sections in Defect, 3D Scaffold, Scaffold + NSs groups at 12 weeks after calvarial defect (bar = 500 μm at low magnification and bar = 50μm at high magnification). **m** Co-immunofluorescent staining of NGFR and OCN expression in Defect, 3D Scaffold, Scaffold + NSs groups at 4 and 12 weeks after calvarial defect (bar = 50 μm), respectively. **n** Immunofluorescent staining of CD31 expression in Defect, 3D Scaffold, Scaffold + NSs groups at 4 and 12 weeks after calvarial defect (bar = 200 μm), respectively. The white dashed line indicates the border between the newly formed bone-like tissues. All data represent mean ± SD, and source data are provided as a Source Data file. At least three times of experiments were repeated independently.

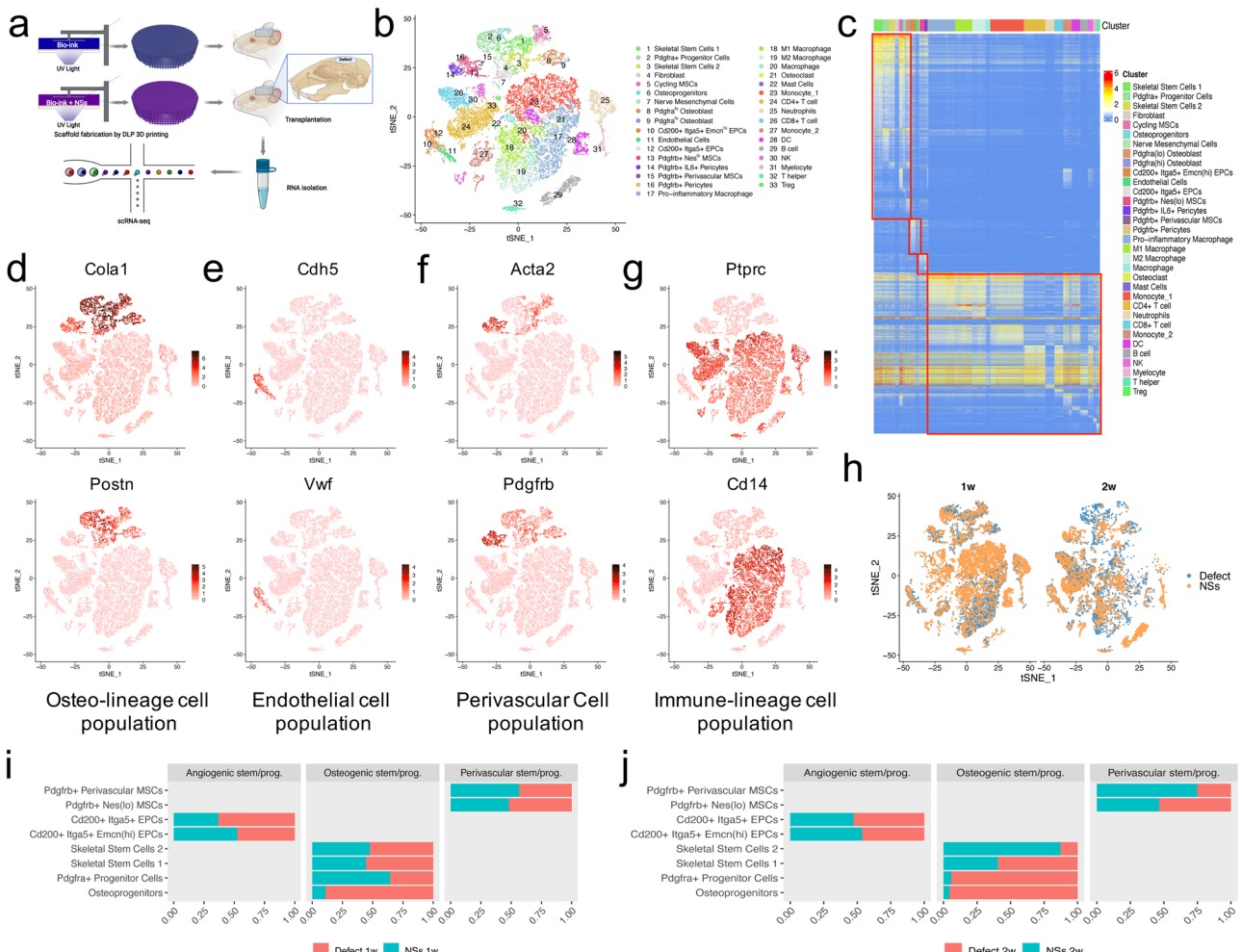

**Fig. 3 | Four cell clusters were differentially regulated by in situ culture system with NSs during bone regeneration. a** Workflow of sample isolation, mRNA libraries preparation, and single-cell RNA sequencing of regenerated tissue from critical-sized calvarial defects with or without the implantation of in situ culture system of NSs-loaded 3D printed hydrogel scaffold, at both 1 week and 2 weeks after defect surgery. $n = 6$ rats per group and per time point (Created with BioRender.com). **b** tSNE plot shows the distribution of 26016 cells that divided into 33 sub-clusters. **c** Heatmap shows expression of marker genes of the 33 sub-clusters. Marker genes were selected according to $p$ value (Wilcoxon Rank Sum test, top 10). **d**–**g** tSNE plots of the expression of specific genes, which mark four different cell populations (osteo-lineage cell population: Col1a1 and Postn; endothelial cell population: Cdh5 and Vwf; perivascular cell population: Acta2 and Pdgfrb; immune-lineage cell population: CD45 and CD14). **h** Distribution of cells from different time and treatments that visualized by tSNE plots. 1w represents 1 week after defect surgery; 2w represents 2 weeks after defect surgery. **i, j** Barplots of the relative proportion of different sub-clusters in three main cell populations between two groups at 1 week and 2 weeks after surgery, respectively.

were mainly four subclusters of mesenchymal stem/progenitor cells. There were obvious features of Fibroblast (negative for Eng and Thy1, Supplementary Fig. 7a, b), Cycling MSCs (positive for Mki67 and Ube2c, Supplementary Fig. 7c, d) and Osteoblasts (positive for Alpl and Mgp, Supplementary Fig. 7e, f). By analyzing of top markers in the remaining four subsets, we found that specific markers distinguished them with well characterized niche factor gene expression (Supplementary Fig. 7g, h). Combined with the featured cell markers that were reported previously[40–43], we annotated these four subsets as: SSC1, SSC2, Adipo-primed Pdgfra⁺ progenitors and Osteo-primed Osteoprogenitors (Fig. 4b, c). These subsets showed distinct GO features (Supplementary Fig. 7i), although they had close spatial relationships with each other in UMAP (Supplementary Fig. 8a–d). In addition, upon NSs delivery, these skeletal stem/progenitor cell subpopulations have different cell-cell interactions with Osteoblasts (Supplementary Fig. 8e, f). These results showed that four skeletal stem/progenitor cells were involved and displayed distinct characteristics in the bone regeneration process.

Since Song Li's group[30] has selectively expanded Pax7⁺ myogenic stem cells from dermal fibroblasts and skeletal muscle stem cells. By scRNA-seq analysis, they revealed that distinct cell subpopulations differently contributed to chemical-induced expansion[30]. We next identified which stem/progenitor cell subpopulations were locally expanded by NSs delivery. Generally, there was a clear increase of Pdgfra⁺ progenitors and SSC2 subpopulations in cell proportion after 1 week and 2 weeks of bone defect, respectively (Fig. 4d, e). The proportion of Pdgfra⁺ progenitors were increased by almost 2-fold in response to NSs when compared with the untreated defect (Fig. 4e). Interestingly, Pdgfra⁺ progenitors that identified in our study showed a pro-adipogenic feature similar to that of mesenchymal stromal cells[42,44], which are responsible for the bone marrow niche maintenance (Fig. 4c and Supplementary Fig. 7h). However, further GO enrichment analysis (Supplementary Fig. 7i) confirmed the primary capability of Pdgfra⁺ mesenchymal progenitor cells in remodeling the extracellular matrix, which favors tissue repair[45]. The results also revealed that Pdgfra⁺ progenitors were related to the "regulation of vasculature development" and "regulation of angiogenesis", which indicates that the regulation for angiogenesis that is required in the early phase of tissue injury repair.

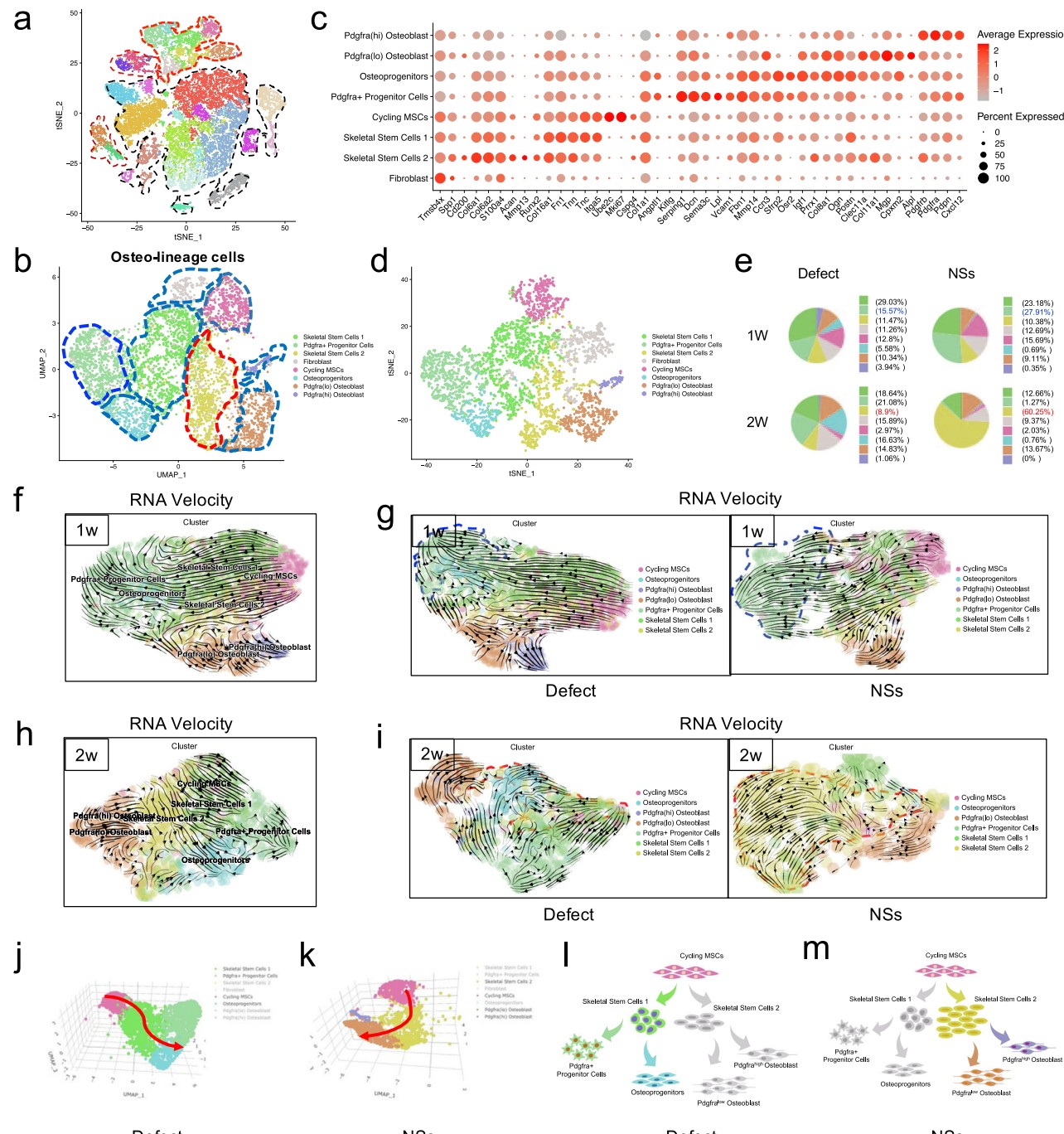

**Fig. 4 | NSs differentially regulated skeletal stem/progenitor cells for osteogenesis. a** tSNE plot shows the distribution of 26016 cells and highlighting of osteolineage cells (red dotted line). **b** Visualization of osteo-lineage cells highlighted in **a** with UMAP plot, which was divided into eight sub-clusters. **c** Dot plots showing the expression of marker genes in the eight osteo-lineage sub-clusters. Dot size represented the proportion of cells expressing specific gene in the indicated subsets and color bar represented the gene expression levels. **d** tSNE plot shows the distribution of cells from the eight osteo-lineage sub-clusters. **e** Relative proportion of cells from Defect and NSs group in each subsets. **f–i** Differentiation trajectory inferred by RNA velocity between Defect and NSs group after 1 week and 2 weeks of defect surgery, respectively. **j, k** Visualization of cells from two different trajectories between Defect and NSs group using 3D UMAP. Red arrow indicates the differentiation direction. **l, m** Schematic diagram of two distinct cell differentiation trajectories between Defect and NSs group.

Surprisingly, 2 weeks after surgery, the cell proportion of SSC2 cells was significantly increased in response to NSs (Fig. 4e). In our data, SSCs can be sub-clustered into SSC1 and SSC2, with common biomarkers of Itga5 and CD200 (Fig. 4c) which indicate a SSC identity[40]. When compared with SSC1, the SSC2 subpopulation showed a more specific expression of Prrx1[29,46], Clec11a[27] and Col11a1[47], which are identified as markers of skeletal stem/progenitor cells. Further GO enrichment analysis confirmed the role of SSC2 in damage restoration

and ECM remodeling after skeletal injury (Supplementary Fig. 7i). Collectively, two distinct skeletal stem/progenitor cell subsets were locally expanded with spatial-temporal specificity upon NSs delivery during the first 2 weeks of bone repair. Moreover, the continuous maintenance and expansion of SSC2 prompted us to explore its contribution to osteogenic differentiation for new bone formation.

To further determine the osteogenic potential of SSC2 upon NSs delivery, we compared the cell fate of the two differentially regulated

stem/progenitor cell lineages. Pseudotime analysis by RNA velocity was performed to explore the lineage relationships among the osteo-lineage cell subsets. The results revealed that Cycling MSCs are situated at the apex of the differentiation hierarchy, and there are strong directional streams from Cycling MSCs toward SSC1 and SSC2, which formed two differentiation trajectories. SSC1 was upstream of both Pdgfra+ progenitors and Osteoprogenitors, while SSC2 was upstream of Osteoblasts (Fig. 4f, h). Further analysis by Monocle also confirmed a similar differentiation continuum spanning SSC1 and SSC2 branches, as well as Osteoblasts and SSC1 downstream clusters of Pdgfra+ progenitors and Osteoprogenitors (Supplementary Fig. 9a). Under the natural healing condition without treatment, Pdgfra+ progenitors and Osteoprogenitors were evenly differentiated from SSC1, and were gradually increased in number as the bone repair proceeded. However, in the NSs healing condition, the Pdgfra+ progenitors were more frequent at the end of the SSC1 branch after the first week of injury, while they were then greatly decreased in number in the following week (Fig. 4f, g). Remarkably, expanding SSC2 was significantly enriched in the differentiation trajectory even after 2 weeks of surgery, and nearly 60% fraction of the osteo-lineage cell subsets were SSC2 (Fig. 4h, i). Although Osteoprogenitors and SSC2 showed osteogenic potential by a similar level expression level of Prrx1, Clec11a and Mgp (Fig. 4c) and a tendency for "Ossification" (Supplementary Fig. 7i), only SSC2 demonstrated a direct differentiation trajectory to Osteoblast. Besides, SSC2 showed a much higher frequency of cell-cell interactions with Osteoblasts than other cell types (Supplementary Fig. 8e, f). More importantly, our data demonstrated that the cells that differentiate along the trajectory of "Cycling MSCs - SSC1 - Pdgfra+ progenitors/Osteoprogenitors" are the main effector cell subsets in an untreated defect (Fig. 4j, l and Supplementary Fig. 9b), and cells that differentiate along the trajectory of "Cycling MSCs-SSC2-sOsteoblasts" trajectory are the additional activated cell subsets that respond to NSs (Fig. 4k, m and Supplementary Fig. 9b).

Taken together, our data indicated that Pdgfra+ progenitors and Osteoprogenitors are the cell sources for regular new bone formation in untreated conditions, while the in situ delivery of NSs dominantly enhanced a new cell branch of SSC subsets by local expansion which is responsible for more efficient bone healing. When compared to the untreated defects, the in situ culture system with NSs exhibits a distinct skeletal stem/progenitor cell atlas during bone regeneration.

## The in situ culture system with NSs induces a local expansion of Msx1+ SSCs for efficient bone regeneration

To better explore the underlying correlation of the NSs and SSC2 expansion, we then focused on the molecular identity of the SSC2 subset. Selected from the osteo-UMAP, the Cycling MSCs, Pdgfra(hi) Osteoblasts and Pdgfra(lo) Osteoblasts each showed separate continuity with SSC2 in a sub-UMAP (Fig. 5a, b), suggesting that the SSC2 subset is one of the important sources for osteogenesis during NSs-induced bone repair. The relatively higher cell number and long-term maintenance of SSC2 (at least for 2 weeks) (Fig. 5c) in the culture system motivated us to investigate the significance of this cell subpopulation. The SSC2 subset exhibited strong expression of various ECM proteins and proteases, such as Col6a5, Col11a1, Ibsp, Acan, and Mmp13 (Fig. 5d), which indicates a role in bone remodeling. We performed GO enrichment analysis on the DEGs of the SSC2 subset, as shown in Fig. 5e, Extracellular matrix organization, Positive regulation of response to external stimulus, Wound healing, response to hypoxia, Ossification were significantly enriched in SSC2. By SCENIC analysis, we identified essential motifs in all osteo-lineage cell subsets, where Runx2 and TCF4 regulons were highly specific in SSC2 (Supplementary Fig. 10a–d). In previous studies, Runx2 and Tcf4 were demonstrated to have profound roles in the transcriptional regulation of osteogenesis[48] and neurogenesis[49], respectively. Interestingly, Runx2/Tcf4-binding

motifs were highly enriched for the co-expressed target genes, Sema4b, Cnp and Sort1 (Supplementary Fig. 10e), which either participate in neurogenesis or as receptors for neurotrophic factors[50–52], while Sort1 could also promote mineralization of the extracellular matrix during osteogenic differentiation[53]. These findings indicate that the activated SSC2 subset transcriptionally responds to NSs and is closely related to osteogenesis and bone healing.

After analysis of the top-ranked cell markers with specificity (Fig. 5f), we found that the SSC2 subset was highly enriched in Msx1, a transcriptional repressor that regulates craniofacial development and odontogenesis[54,55]. In the developing skull, Msx1 is expressed in the suture mesenchyme and dura mater and extends into the postnatal stages of skull morphogenesis[56]. However, in our data, Msx1 also showed a partial enrichment in the Osteoblasts subsets (Supplementary Fig. 11a, b). Feature plot analysis showed that separate regions of Msx1+ cells can be labeled with different cell markers, and Msx1+Alpl+ cells mark the Osteoblasts (Supplementary Fig. 11c–e), suggesting that Msx1+ cells contain subpopulations at multiple states linked to osteogenesis. To elucidate the link between SSC2 and Osteoblasts, we used Monocle 2 unsupervised pseudotime analysis[57] to model the relationship between these two cells, which revealed three sub-clusters (Supplementary Fig. 11f). There are two Msx1+ clusters (cluster 1 and 2) linked to osteoblast fate (cluster 3). Cluster 1 represents Msx1+Acan+ cells and cluster 2 represents Msx1+Col6a5+ cells, and cluster 2 showed a trend toward cluster 3 (Supplementary Fig. 11g–j). Pseudotime projection from cluster 1 to cluster 3 indicates that only one single cell marker cannot precisely define the SSC2 subset during the regenerative process.

We then analyzed specific gene expression in different subsets and identified that the top markers Mmp13 and Edil3 were specifically expressed in the SSC2 (Fig. 5f). As Mmp13 is a well-known gene required for normal embryonic bone development and ossification[58], we re-examined the SSC2 dataset and found that Msx1+ cells that co-express Mmp13 could define SSC2 more accurately. A co-localization of Msx1 and Mmp13 by visualization analysis showed that Mmp13+ cells overlapped with Msx1+ cells and were mainly located within the SSC2 region (Supplementary Fig. 12a–c), which suggests that the Msx1+ Mmp13+ double-positive makers were more specific to the SSC2 and did not include Osteoblasts. To validate the scRNA-Seq analysis in vivo, we checked the co-expression pattern of Msx1 and Mmp13 using immunofluorescence staining. Compared to an undetectable level in the untreated defect, Msx1 was found to be widely expressed both in the upper and middle layers of the gel scaffold loaded with NSs after 2 weeks of injury. Mmp13 was also detected and partially co-localized with the Msx1 expression in the regions where only SSC2 was locally expanded (Fig. 5g). To more directly verify the effect of NSs in expanding SSCs in vitro, we isolated and cultured the fresh primary bone marrow tissue (a known source of SSCs) on the cell culture gel system with or without NSs (Supplementary Fig. 12d). After 2 weeks of expansion in vitro, the overall number of Msx1+ cells was significantly increased in response to NSs, and almost half of the Msx1+ cells were also Mmp13+, suggesting a local expansion of SSC2 (Supplementary Fig. 12e). CD200, as a cell surface marker of mouse skeletal stem cells[40], is also specifically expressed in the SSC2 subpopulation in our data (Supplementary Fig. 12f). Through flow-cytometric analysis of markers against Msx1 and CD200, we observed that the Msx1+ CD200+ cells were increased to 13.1% after 2 weeks of NSs-loaded scaffold culture, when compared to the vehicle scaffold culture (Supplementary Fig. 12g, h).

Together, the Msx1+ SSC2 subset transcriptionally responds to NSs and is responsible for in vivo osteogenesis. Acting fundamentally as injury-responsive skeletal stem/progenitor cells at base level, Msx1+ SSC2 were able to be locally expanded by the in situ culture system with NSs for more efficient bone repair.

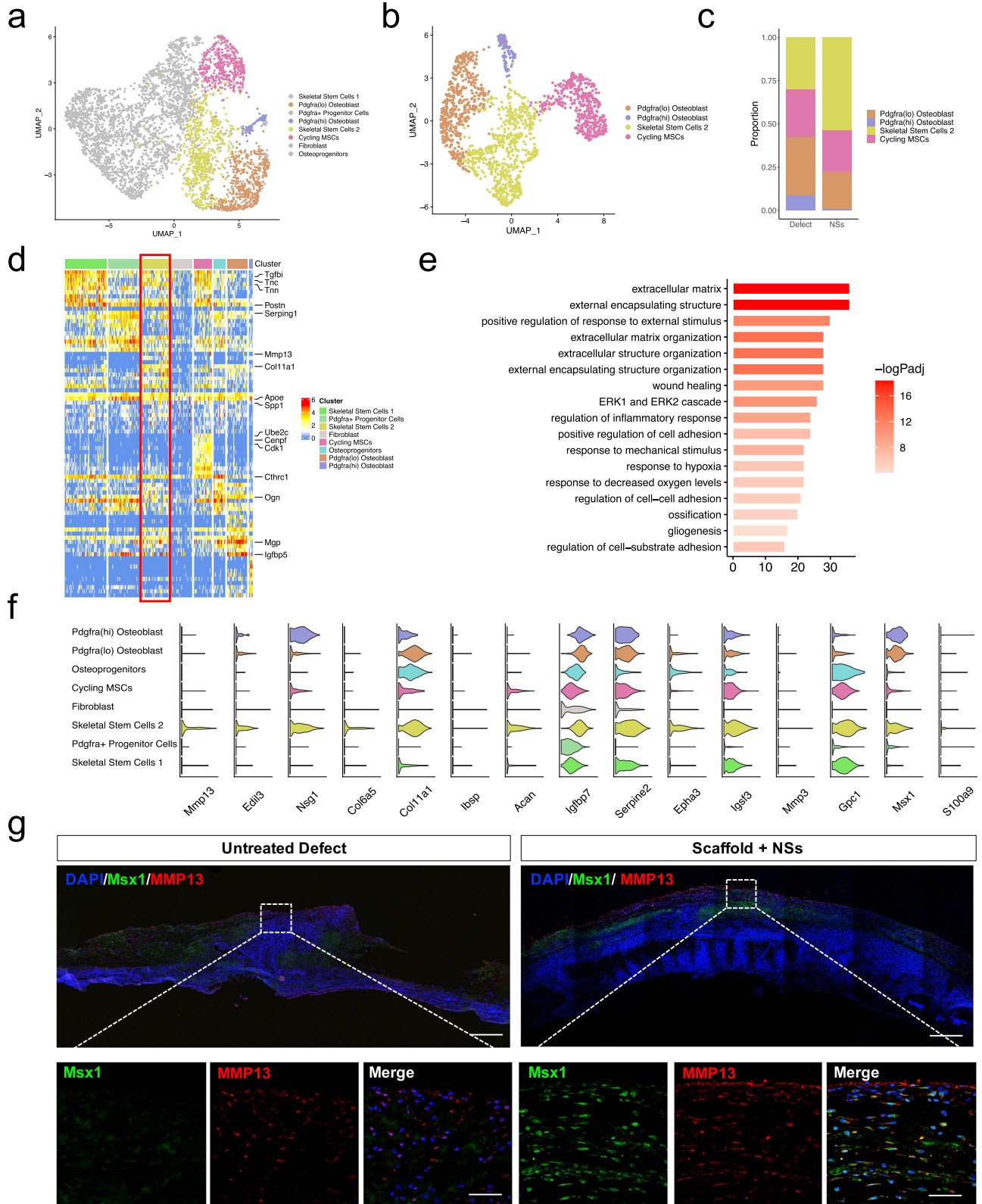

**Fig. 5 | Msx1⁺ SSCs subset was locally expanded by the in situ culture system with NSs.** **a** Visualization of SSC2 differentiation lineage cells in osteo-lineage cells with UMAP plot, highlighting four specific subsets. **b** UMAP plot of the four specific subsets. **c** Relative proportion of cell subsets between Defect and NSs group. **d** Heatmap and the expression of marker genes of 8 osteo-lineage sub-clusters. Marker genes are provided in source data. **e** Barplot showing the GO enrichment in SSC2 subset. Markers used for enrichment analysis were selected according to *p* value (\**p* < 0.05, Wilcoxon Rank Sum test) and fold change (>1).

Color bar represented the adjusted *p* values performed by R package cluster-Profiler (Benjamini–Hochberg). **f** Violin plot showing the SSC2 top marker gene expression levels across the whole 8 different subsets. **g** Co-immunostaining of Msx1 and Mmp13 expression of paraffin sections in Untreated Defect group and Scaffold + NSs group at 2 weeks after defect surgery (bar = 200 μm at low magnification and bar = 30 μm at high magnification), at least three times of experiments were repeated independently.

## The in situ culture system with NSs promotes rapid bone regeneration partially through endochondral ossification

We then wondered how the locally expanded Msx1⁺ SSC2 subset contributed to the rapid bone formation, since 2 weeks after injury is still in the early stage of osteogenesis. To further determine the difference in osteogenic patterns between the pro-regenerative NSs delivery and untreated condition, we reconstructed the cell differentiation trajectory by the Monocle 2 R package. We found one major branchpoint that diverted cell fate from Cycling MSCs toward SSC1-downstream and SSC2-downstream lineages (Fig. 6a). Consistent with the RNA velocity analysis, SSC1 and SSC2 were both located at the branching point of the bifurcation. When Cycling MSCs were set as the root to identify temporally expressed genes over pseudotime, we found that genes were highly expressed in SSC2 (e.g., Serping2, Fxyd5, Col11a1) were gradually down-regulated, while genes that highly expressed in Pdgfra+ progenitors or Osteoprogenitors (e.g., Gsn, Ifi27l2b, Mnda, Ackr3, Sfrp2) and Osteoblast (e.g., Ptgds, Bglap, Igf2) were upregulated upon terminal differentiation (Fig. 6b). Interestingly, we found that cell distributions along the differentiation trajectory follow time-dependent patterns regardless of the treatment during regeneration, and the branchpoint of cell differentiation basically appeared after the first 1 week of injury (Fig. 6c). As Msx1⁺ Mmp13⁺ cells were activated by NSs at all observed time period periods of bone repair in our study (Fig. 5h and Supplementary Fig. 13a), we examined the specific cell marker expression of SSC2 in the pseuo-timeline. In our data, we found that Msx1 was extensively expressed across the whole timeline, while Acan and Col6a5/Mmp13 showed specific expression in the early and late differentiation stages, respectively (Fig. 6d). Thus, we speculated that Msx1⁺ SSC2 may dynamically evolve in a time-dependent manner with variable gene expression. In featureplot analysis, Msx1⁺ SSC2 showed two distinct stages of marker gene expression, which verified the results above (Fig. 6e). Besides, during bone repair, the upregulated Acan expression in the early stage and Mmp13 expression in the late stage strongly suggested the association of Msx1⁺ SSC2 in endochondral ossification[59].

To validate the time-dependent appearance of Msx1⁺ SSC2 in vivo, we used immunofluorescent staining for confirmation. Consistent with the pseuotime analysis, Msx1⁺ cells mainly resided in the surface or inner surface of the NSs-contained scaffold at both time points, and the co-expressed Acan was specifically appeared at 1 week after surgery (Fig. 6f), while the co-expressed Mmp13 was specifically appeared at 2 weeks after surgery (Fig. 6g). These results indicated that the locally expanded Msx1⁺ SSC2 promotes bone regeneration may partly through an endochondral pathway. As the chondrocyte phenotype can only be observed as early as 2 weeks after chondrogenic induction initiates[60], in our data, cells migrated into the in situ culture gel scaffold and aggregated in the voids between hydrogel grids at 1 week after injury (Fig. 6h). Under the treatment of delivered NSs, the sections harvested from the gel scaffold after 2 weeks were found to retain a cartilage-like tissue with rich proteoglycan staining (Fig. 6i) and COL2 labeling (Supplementary Fig. 5e), and became hypertrophic cartilage (COL10 expression, Supplementary Fig. 5f) until new bone formation after 12 weeks (Supplementary Fig. 13b, c). In addition, the vascular invasion that needed for further endochondral ossification was also revealed by staining for the endothelial marker, CD31. Along with the enhanced chondrogenesis, the cells in the culture gel scaffold with NSs were surrounded by a more abundant population of CD31⁺ cell distribution (Supplementary Fig. 5e), and eventually mixed with each other eventually (Supplementary Fig. 13c).

Based on the scRNA-seq data, we found that Prx1, which is a marker for the limb-bud mesenchyme[29] and an upstream skeletal stem cell marker[61], was almost expressed in all the osteo-lineage cell and perivascular cell clusters (Supplementary Fig. 12b). Meanwhile, the Msx1 expression was restricted in the osteo-lineage cell clusters (Supplementary Fig. 12a). To further determine the role of Msx1⁺ SSC2 subset in osteogenesis and endochondral bone formation, the in vivo lineage potential was assessed by generating mice of the genotype Prx1-Cre; Rosa26-tdTomato to report lineage contribution (Fig. 7a). Prx1-lineage-marked cells with a partial overlap expression of Msx1 are observed throughout the newly formed tissues in skull bone with NSs-loaded scaffold both 2 weeks and 4 weeks after injury (Fig. 7b–g), demonstrating the local expanded Msx1⁺ SSC2 subset was largely restricted to the Prx1-lineage cells. Two weeks after surgery, most of the Msx1⁺ cells at the upper layer of the regenerated tissues showed positive expression of RUNX2 and again showed overlap with the ACAN-positive cells, which indicates the initiation of osteogenesis, as well as the possible chondrogenesis (Fig. 7d, f). Following a 4-week chase, the Prx1-lineage-marked cells with Msx1 expression eventually contributed to the OCN-positive osteoblasts (Fig. 7e). However, the contribution of mature COL2-expressing chondrocytes were not observed in the Prx1-lineage-derived Msx1⁺ cells at this stage (Fig. 7g). Of note, the existed COL2-postive cells are still co-expressed with Msx1, consistent with the differentiation potential of Msx1-expressing population in the early regenerative stages of rat skull model. Different from the development stage, the Msx1⁺ SSCs subset is one of the main source for high efficient osteogenesis and closely correlates with the chondrogenesis.

Collectively, these findings indicated that the locally expanded Msx1⁺ SSC2 subset by the in situ culture system with NSs contributed to the enhanced bone regeneration. The differentiation trajectory showed a molecular feature of Msx1⁺ SSC2 that underwent the endochondral pathway in this study, together with lineage-tracing results, our findings suggest that Msx1⁺ SSC2 acquired an endochondral bone formation capacity for injury repair, rather than a direct intramembranous ossification.

## Discussion

Full-thickness regeneration of critical-sized calvarial defects was successfully achieved by implantation of the in situ cell culture system consisted of a 3D-printed hydrogel scaffold incorporated with neurotrophic supplements (NSs). Notably, by single-cell RNA Seq analysis, we found that the in situ culture system with NSs locally expanded the Msx1⁺ Skeletal Stem Cell (SSC) subpopulation that promoted efficient bone healing through an endochondral ossification pathway. Our findings on the expansion of in situ skeletal stem cells via a specific tissue-engineered graft during the bone regenerative process could provide a rational cellular basis for the efficient regeneration of large bone defects. This NSs-containing in situ cell culture system is highly biocompatible and reproducible with a standard production process. In situ expansion of resident SSCs is promising for highly efficient regeneration of critical-sized bone defects and holds potential for clinical applications.

Previous studies have performed extensive investigation of biomedical engineering approaches using biomaterials and stem cell transplantation for bone regeneration. However, the full-thickness defects of the calvarium still present reconstructive challenges. Stem cell-based therapies are often hindered by inadequate in vitro expansion efficiency, long culture times, and limited in vivo survival rates. In addition, after several passages of culture, the multi-lineage commitment of the heterogeneous stem cells cannot be precisely controlled after implantation. To avoid these difficulties, in the present study, we focused on the activation of endogenous stem cells by in situ delivery of soluble factors to regenerate bone tissues. After injuries, the recruited endogenous stem cells migrate into the lesion and participate in tissue regeneration. Upon fractures or defects, the periosteum and bone marrow are essential sources of resident stem cells for bone repair. However, the self-renewal ability and transcriptomic diversity of skeletal stem cells is often diminished in adult bone tissues and even exhausted in aged individuals[28]. Recent studies from leading research groups in this area have shown that, by co-delivery of soluble rhBMP-2

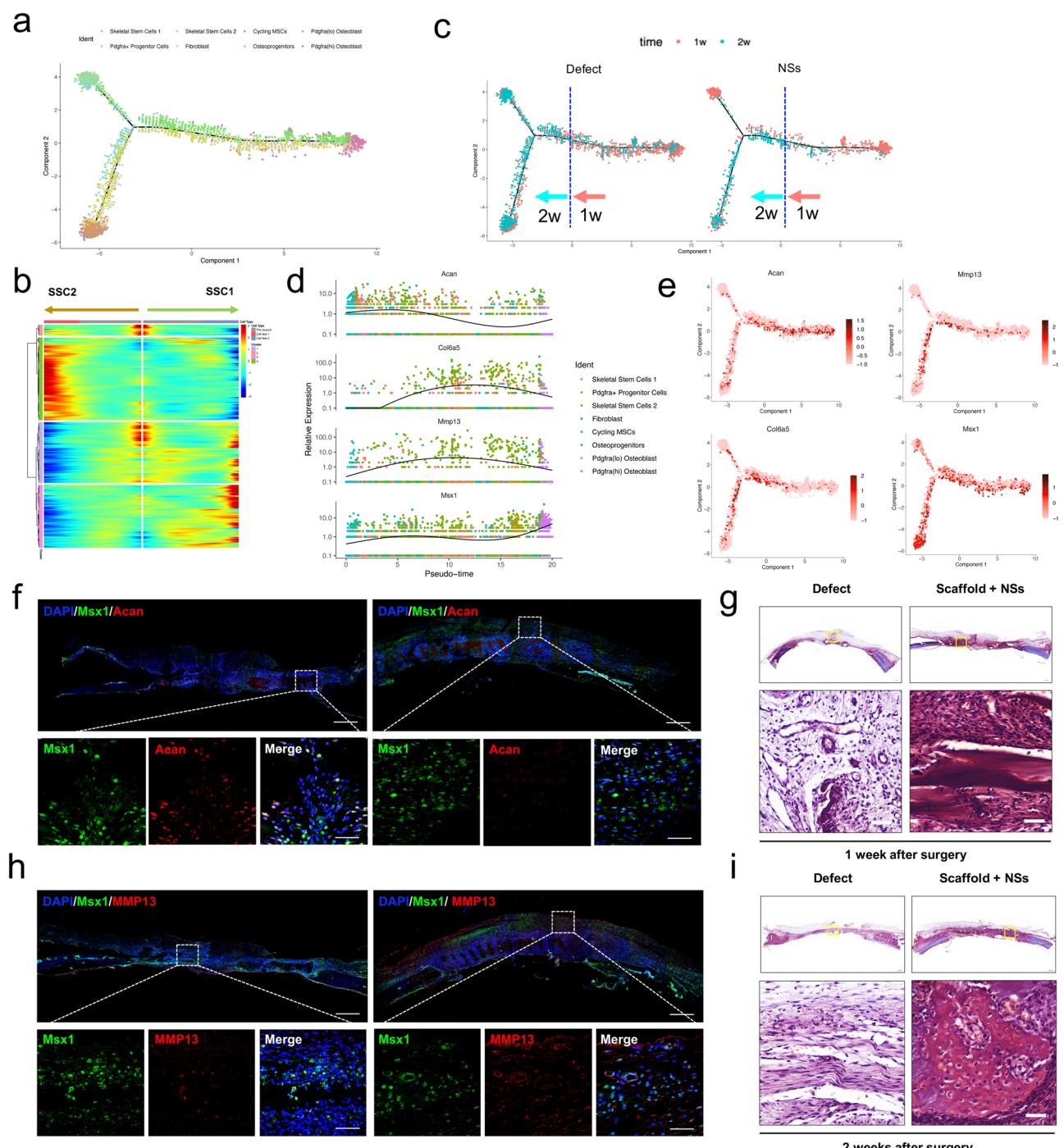

**Fig. 6 | In situ expansion of Msx1⁺ SSCs subset promoted efficient bone regeneration partially through endochondral ossification. a** Trajectory of differentiation from cycling MSCs to both SSC1 and SSC2 lineages predicted by Monocle 2. **b** Heatmap of gene expressions in subsets ordered by pseudotime of two differentiation trajectories in **a**. **c** Distribution of cells on both the differentiation trajectories from Defect and NSs groups showing a featured change dominated at 1 week and 2 weeks after surgery, respectively. **d** Relative expression level of anabolic and metabolic genes (Acan, Col6a5, Mmp13) of cartilage and Msx1 gene along the whole pseudotime. **e** Expression of the above chondrocyte-specific genes (Acan, Col6a5, Mmp13) and SSC2 marker gene (Msx1) visualized on differentiation trajectory. **f** Co-immunostaining of Msx1 and Acan expression of paraffin sections in Scaffold + NSs group at 1 week and 2 weeks after defect surgery, respectively (bar = 200 μm at low magnification and bar=30μm at high magnification). **g** Co-immunostaining of Msx1 and Mmp13 expression of paraffin sections in Scaffold + NSs group at 1 week and 2 weeks after defect surgery, respectively (bar = 200 μm at low magnification and bar = 30 μm at high magnification). **h**, **i** Safranin-O staining of paraffin sections in Untreated Defect group and Scaffold + NSs group at 1 week and 2 weeks after defect surgery, respectively (bar = 500 μm at low magnification and bar=50μm at high magnification). At least three times of experiments were repeated independently.

and VEGF receptor antagonist, lineage-specific induction of resident skeletal stem cell populations efficiently induced cartilage regeneration rather than fibrous tissue formation[62]. Combinatorial treatment with rhBMP2 and a CSF1 antagonist reactivated aged skeletal stem cells

in vivo and promoted bone regeneration to a youthful level[28]. Nevertheless, due to short-lived activated stem cells, current approaches by promoting the differentiation potential are still inefficient for full-thickness regeneration of critical-sized bone defects. In our results, we

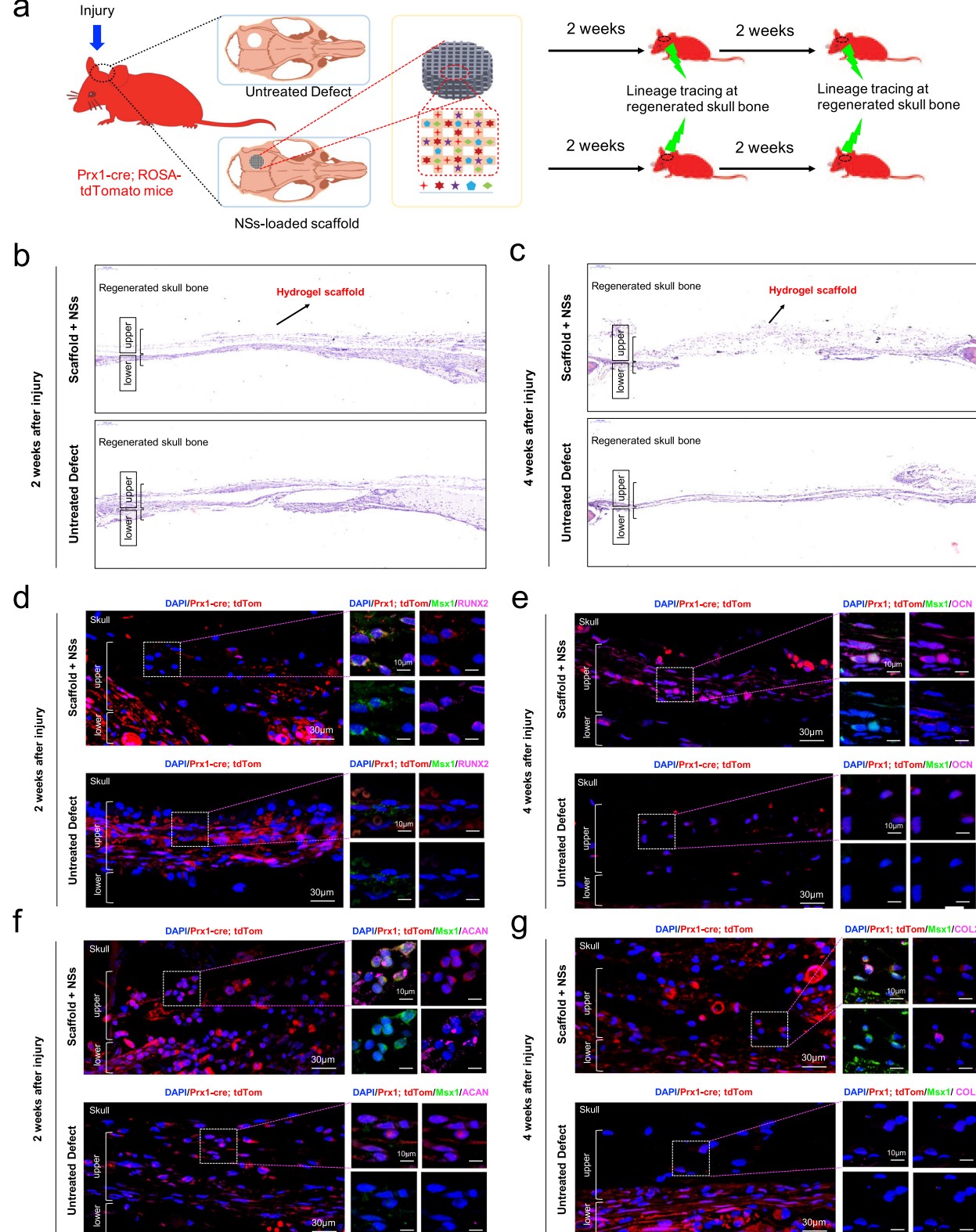

**Fig. 7 | Msx1⁺ SSCs subset is a source of osteogenesis and closely correlates with endochondral ossification. a** Schematic of the assessment of Msx1⁺ SSCs subset in differentiation hierarchy by Prx1-lineage tracing mice. **b, c** HE staining of newly formed calvarium tissues in skull bone injury at 2 weeks and 4 weeks after surgery, respectively. **d–g** Coronal sections of the calvarium tissues from Prx1^Cre; Rosa^tdTomato mice co-immunostained with Msx1 and Runx2 (**d**); Msx1 and OCN (**e**); Msx1 and ACAN (**f**); Msx1 and COL2 (**g**); Scale bars: low magnification: 30 μm, high magnification: 10 μm. At least three times of experiments were repeated independently.

found that the in situ cell culture system with NSs promoted rapid bone regeneration through a locally expanded resident skeletal stem cell (SSC) subpopulation. Therefore, we conclude that the enhanced lineage differentiation of SSCs and rejuvenation of the aged skeletal niche, as well as the in situ expansion of SSCs are particularly important for efficient skeletal tissue regeneration.

SSC subpopulations appear during bone development and postnatal growth, and play important roles in bone maintenance and repair in response to injuries. Using lineage-tracing techniques, a large number of studies have revealed several skeletal stem/progenitor cell populations. In contrast to the Lepr which is a hall-marker of bone marrow MSCs and labels the main source of bone-forming cells throughout the marrow cavity[24,27], Gremlin 1 identifies a rare skeletal stem cells underneath the growth plate that contribute to the formation of osteoblast, chondrocytes, and reticular marrow stromal cells in postnatal and impaired skeletogenesis[23]. In addition, Gli1+ marks skeletal progenitor cells residing in the bone marrow niche and are responsible for the progressive developmental bone formation and fracture repair[63]. Thus, these studies highlighted the heterogeneous identities and osteogenic potential of the SSCs upon bone injuries. Subsequently, Michael T. Longaker & Charles K. F. Chan's group have made great progress in identifying SSCs in mice[40] and human[64]. The CD45⁻Ter119⁻AlphaV⁺Thy⁻6C3⁻CD105⁻CD200⁺ subpopulation that was isolated by fluorescence activated cell sorting (FACS) from mouse bone tissues was identified as mSSCs, which shows a skeletal lineage-restricted feature and possesses stem-cell-like characteristics of self-renewal and multipotent differentiation into bone, cartilage and stroma but not fat both in vitro and in vivo[40]. Similarly, by single-cell RNA sequencing analysis, human SSCs from the growth plate and diaphysis were also identified by a combination of cell surface markers, PDPN⁺CD146⁻CD73⁺CD164⁺, which are activated to undergo local expansion following skeletal injury[64]. Therefore, skeletal stem cells can be marked and purified by defined cell markers from skeletal tissues, and there are evolutionary differences in their identity among various species. The above definitions for SSCs may differ from our findings referring to skeletal stem/progenitor cells from rat species, but our results did show a more closer relationship to cell markers from mice. Nevertheless, there are still limitations in our annotation attempts for SSCs, which need to be further characterized in the future work.

Interestingly, the increased SSCs that responsible for the improved tissue regeneration were due to a local expansion, rather than recruitment[65], suggesting that there is a practical feasibility for the in situ delivery of soluble factors with pro-proliferative properties. In fact, SSCs are heterogeneous and required for skeletal development, homeostasis maintenance, and injury repair throughout life[23,40,64,66]. In addition, specific subpopulations of endogenous stem cells are usually the main regenerative cell types for bone regeneration[67,68], and previous studies have shown unique potential for cell expansion by defined xeno-free medium[29,30]. Our results showed that neurotrophic factors-supplemented growth medium have greatly promoted the cell proliferation of MSCs in both 2D culture and the hydrogel scaffold system. Moreover, those expanded MSCs activated by NSs treatment showed a significantly higher potential for osteogenic differentiation (Fig. 1e, i), strongly indicating the activation of stem cells at the subpopulation level. Further experiments of bone marrow culture on the NSs-loaded hydrogel scaffold system also showed a significant expansion of the Msx1⁺Mmp13⁺ cell subpopulation (Fig. 5j). Since specific culture medium generates expansion for specific stem cell subpopulations, the results from our study suggested that it is available to induce a local expansion of stem cell subpopulations by soluble factors with known functions.

More recent studies have revealed the presence of SSCs in a different anatomical location, the periosteum[68,69]. Besides, periosteum-resident SSCs that specialized in periosteal physiology were found in both the long bones and calvarium[41]. Although bone marrow and periosteum are both important sources for SSCs, periosteal SSCs that undergo intramembranous bone formation at baseline and contribute to the endochondral process after fracture are the major source of endogenous stem cells for cortical bone repair[41,69]. In addition, suture stem cells have long been recognized as osteogenic mesenchymal stem cells for craniofacial bone homeostasis and repair[70], and coincidentally, periosteal SSCs are present in the calvarial suture[41,71]. Among Hox homeobox genes, Msx1 (Hox7) was found expressed specifically in the suture mesenchyme and plays an early morphogenetic role during craniofacial bone development[56,72] and drives osteoblast differentiation[73], while Hox11 is determinant for patterning forelimb zeugopod skeleton and Hox11-lineage skeletal stem cell from bone marrow and periosteum give rise to all skeletal lineages throughout the life[67]. Although previous studies have shown that Msx1 expression extends into the postnatal stages of skull morphogenesis and is surprisingly involved in the active membranous and endochondral bone formation in skull bone and beyond[72], so far, none of the studies have illustrated the role of Msx1-expressing cells for skull bone defect regeneration. In our results, Msx1⁺ SSC2 that found in regenerated tissue of skull bone injury was demonstrated to be the progenitor cells of osteoblasts and chondrocytes (Figs. 4 and 7), further confirmed the high efficient bone-forming potential driven by Msx1.

Although and multiple biomedical engineering approaches have been developed for more efficient bone repair, very few have clarified the cellular mechanism at the single-cell level during regeneration. To address the mechanisms that are responsible for the full-thickness regeneration of calvarial defects by the in situ delivery of NSs in our study, we used single-cell RNA-Seq technology to analyze the in vivo cell atlas regarding the NSs-enhanced healing. We demonstrated that implantation of the culture system greatly enhanced the in situ recruitment and expansion of a specific Msx1⁺ SSCs subset, which shows a much closer link to osteoblast formation and acquired a distinct differentiation trajectory during injury repair, as shown by the pseudotime analysis. We also found that the enhanced calvarial regeneration was accompanied with an endochondral ossification that was closely correlated with Msx1⁺ SSCs, even though the calvarial bone is a well-known site of intramembranous ossification.

Apart from the well-defined Msx1⁺ SSC subpopulation in this study, we also clarified the biological significance of the overall altered SSCs subpopulations that were greatly expanded by NSs delivery. We carefully examined the characteristic identities and dynamic changes of a SSC1 downstream Pdgfra⁺ Progenitors during the bone regeneration. Actually, previous studies have shown that Pdgfra⁺ MSCs are responsible for the bone marrow niche maintenance, and Pdgfra⁺ mesenchymal stromal cells within a perivascular niche of skeletal tissues[46] play a dual role in promoting both revascularization in the early repair stage and pathological fibrosis if they persist as differentiated cells[45]. Coincidentally, the relative cell proportion of Pdgfra⁺ progenitors was diminished to a negligible level after the second week post-injury (Fig. 4e). The dynamic cellular changes suggested that pro-regeneration rather than pro-fibrosis was favored in the culture system with NSs, as persistent fibrosis is usually the obstacle for bone healing[61]. Despite of the diversity of tissue origins, SSCs that are maintained within different bone tissues can be reactivated and locally expanded in response to injuries for more efficient fracture repair[41,65]. In contrast to those studies, our findings revealed two locally expanded skeletal stem/progenitor cells with spatially distinct characteristics that existed among the two different trajectories during bone repair. Together, our discoveries suggest that there is a rational basis for full-thickness reconstruction of critical-sized calvarial defects, which is still in need of further investigation.

Another highlight of this study is the development of a 3D-printed hydrogel scaffold as a carrier for local delivery of soluble factors and

in situ stem cell expansion. With the development of microfabrication technology, the synthesis of biomimetic nano-hydroxyapatite[74] and hierarchical structure design or mineralized coatings have emerged[75] that promoted osteogenic differentiation and rapid collagen mineralization, respectively. However, due to a limited degradation, tissue integration, and the space-occupying effect of synthetic grafts, the new bone formation cannot completely replace the scaffold to fill the defect, making it difficult to regenerate completely[76]. In addition, implanted scaffolds obtained from synthetic materials often result in an uneven distribution of endogenous cells and a limited cell survival rate[77,78] in vivo. Our results showed that the photo-crosslinked 3D-printed hydrogel scaffold with neurotrophic supplements effectively enhanced the proliferation, adhesion, migration and osteogenic differentiation of mesenchymal stem cells in vitro and in vivo, and degraded within 4 weeks of implantation. Although hydrogels provide an available delivery system for soluble factors, when using hydrogel scaffolds for bone regeneration, there are obvious limitations in mechanical support. Therefore, the histological and functional assessment of the in situ hydrogel culture system was largely restricted to calvarial defects, rather than a long bone segmental defect model.

In summary, our study indicates that inducing SSCs and their local expansion with potent soluble factors could be a promising strategy for bone regeneration. The implementation of this study clarified the skeletal stem cell subpopulations and their osteogenic regulation mechanism in the bone healing process and provided a single-cell level theory for the rapid regeneration of large bone defects.

## Methods

### Human bone marrow-derived stem cells (BMSCs) isolation and culture

Human BM aspirates were obtained in The Second Affiliated Hospital of Zhejiang University (Zhejiang University, China) with written informed consent from orthopedic individuals with femoral fracture (Male, 2 donors, an average age of 37). All samples were obtained and used according to standard guidelines approved by the Zhejiang University Ethics Committee (Ethical NO. ZJU2016003). The samples were processed for the following experiments, after the filters used to trap bone spicules and cell aggregates, and were carefully and aseptically washed with cold PBS several times. BMSCs were isolated using a Percoll gradient (Gibco) and cultured in Low-Glucose Dulbecco's Modified Eagle's Medium (DMEM; Gibco) containing 10% fetal bovine serum (FBS; Gibco), 1% penicillin/streptomycin (P/S; Thermo Fisher Scientific) and incubated at 37 °C, 5% $CO_2$. The culture medium was replaced every other day until about 90% confluence was achieved. Passage 3-5 of the BMSCs were utilized for the in vitro experiments either in 2D or 3D hydrogel scaffold system in this study.

### Rat bone marrow isolation and seeding on hydrogel scaffold

The bone marrow from the rat femora and tibia was flushed in freshly prepared culture medium with L-DMEM medium (DMEM; Gibco), 10% fetal bovine serum (FBS; Gibco), and 1% penicillin/streptomycin (P/S; Thermo Fisher Scientific). After centrifuged, the bone marrow was resuspended in growth medium and then seeded on the hydrogel scaffold, and samples were analyzed in certain time points.

### Neurotrophic Supplements

The NSs medium contained 2% StemPro® Neural Supplement (Gibco) and 1% B27 supplements (Invitrogen); recombinant human fibroblast growth factor (FGF)-basic (bFGF, 20 ng/ml), recombinant human epidermal growth factor (EGF, 20 ng/ml), GlutaMAX™-I Supplement (1%, Gibco), in Dulbecco's modified Eagle's medium (DMEM)/ F12 (1:1) containing 10% fetal bovine serum (FBS, Gibco) and 1% antibiotics.

### Cell viability assay

$1 \times 10^5$ cells/well were suspended in DMEM media supplemented with 10% fetal bovine serum (FBS) and 1% penicillin/streptomycin with or without NSs. And the cells and media were incubated for 1, 3, 5, 7 days and the CCK-8 kit (Dojindo, Japan) was chosen to determine cell viability.

### Flow cytometry analysis of expanded mesenchymal stem cells

For flow cytometry analysis, followed by dissociation of the expanded MSCs, $1 \times 10^5$ cells were suspended in 100 µl of 2% FBS in PBS and incubated with primary antibody (diluted 200×) (CD45-PE-CyTM7 (368510, Biolegend, San Diego, CA, USA); CD90-APC (328114, Biolegend, San Diego, CA, USA); CD29-APC (17-0299-42, Invitrogen, USA); CD105-PE (85-12-1057-42, eBioscience, San Diego, CA, USA); and CD200 (0.25 µg/10⁶ cells, AF2724, R&D Systems, USA); Msx1 (diluted 200×, NBP2-57969, NOVUS, USA).) for 1 h on ice. The cells were then washed and suspended in 100 µl of 2% FBS in PBS containing secondary antibody (diluted 1000×) for 30 min on ice. Fluorescence was detected using a CytoFLEX S flow cytometer (Beckman Coulter) and data were analyzed using the FlowJo software. Gating strategy used for analyses is available in Supplementary Fig. 14.

### Calculation of cell Population Doubling Time (PDT)

In vitro expanded cells were harvested and then counted. PDT of the cells from P2 to P4 was calculated by the following calculation formula: PDT = Culture Time * log(2)/[log(Final Cell Numbers) − log(Initial Cell Numbers)].

### CFU-F assay

For CFU-F cultures, expanded MSCs were seeded in 6-well plate (100 cells/well) containing culture medium (L-DMEM supplemented with 10% FBS, 1% Penicillin/Streptomycin) and incubated at 37 °C with 5% $CO_2$. Half of the medium was changed every other day. At day 10, cells were fixed and stained with crystal violet staining solution, and adherent colonies with more than 50 cells were quantified.

### Multilineage differentiation assays

For adipogenic differentiation, the confluent cells were cultured with adipogenic medium containing α-DMEM with 10% FBS supplemented with 10 µg/ml insulin, 100 µM indomethacin, 0.5 mM 3-iso- butyl-1-methylxanthine, and 0.1 µM dexamethasone (Sigma, St. Louis, MO). The medium was changed every 2 days during 2 weeks. For osteogenic differentiation, the cells were plated at confluence in osteogenic medium containing α-DMEM with 10% FBS supplemented with 0.1 µM dexamethasone, 0.2 mM L-ascorbic acid, and 10 mM glycerol 2-phosphate disodium salt hydrate (Sigma, St. Louis, MO). The medium was changed every 2 days during 1 or 2 weeks. For chondrogenic differentiation, the cells were resuspended at a concentration of $5 \times 10^5$ cells per mL in chondrogenic medium consisting of high-glucose DMEM, 1% ITS(Gibco), 100 nM dexamethasone (DEX, Sigma), 1% P/S, 10 ng/ml TGF-β3 (Novoprotein), 50 µg/ml L-ascorbic acid 2-phosphate (ascorbate), 40 µg/ml L-Proline (proline), and 100 µg/ml sodium pyruvate (Gibco). Cells were then centrifuged for 5 min at 300 ×g to form a pellet. The medium was changed every 2 days during 3 weeks. Chondrogenic pellets were cultured at 37 °C for up to the timepoints required for various experiments.

### Oil Red O Staining

After Medium was removed, and cells were washed with PBS and fixed with 4% PFA for 30 min at room temperature. Cells were rinsed again with PBS and stained for 30 min with Oil Red O working solution (3:2 dilution with distilled water). Cells were then observed under a light microscope after 4−5 times of washes with PBS.

## Alkaline Phosphatase (ALP) activity

BMSCs were seeded in 24-well plate at a density of $2 \times 10^4$ cells/well, and 24 h post-seeding the medium was changed to experimental medium. Cells were briefly washed with PBS and fixed for 10 min with 2% paraformaldehyde in PBS (Sigma). Cells were washed twice with PBS and incubated for 20 min at room temperature with 50 mg/ml Naphthol AS-MX phosphate, 0.5% N,N-Dimethylformamide, and 0.6 mg/ml Fast Red Violet LB in 0.1 M Tris-HCl, pH 8.9.

## Alizarin Red Staining

Cells were washed with cold PBS and fixed with 70% ethanol for 15 min on ice. Cells were washed with distilled water and stained with 2% alizarin red solution for 5 min. Cells were then washed thoroughly with distilled water and air dried before microscopic visualization.

## Alcian Blue Staining

After medium was removed, and cultures were washed with PBS and fixed with 4% PFA for 30 min. Cultures were stained for 30 min with 1% Alcian Blue solution 8GX (Sigma) in 3% acetic acid, pH 2.5, and washed three times with 0.1 N HCl and then with PBS.

## Synthesis of GelMA

GelMA (Methylacrylated Gelatin) was synthesized according to a previous study[37]. Briefly, type A gelatin (Sigma-Aldrich) was dissolved in PBS at 50 °C and stirred to make a 10% w/v homogeneous solution. And a 0.1 mL methacrylic anhydride (MA) (Sigma-Aldrich) per gram of gelatin was added to homogeneous gelation solution at a rate of 0.5 mL per minute with a continuous stirring. The mixed solution was allowed to react at 50 °C for 3 h with stirring. Then, the GelMA solution was poured into 8–14 kDa cutoff dialysis tubing (VWR Scientific USA) and dialyzed against deionized water for 6 days at 50 °C to remove untreated MA and other byproducts. The deionized water was replaced every 1–2 days. The resulted GelMA solution was frozen overnight or longer at −80 °C and lyophilized and stored at −20 °C for further use.

## Synthesis of NB and HA-NB

Methyl 4-(4-(hydroxymethyl)−2-methoxy-5-nitrophenoxy) butanoate (mNB) was synthesized based on a previous research[79]. mNB (0.5 g, 1.8 mmol) and ethylenediamine (1.1 mL, 2 mmol, Sigma Aldrich) were dissolved in methanol. Then, the mixture was refluxed overnight until the starting individual components were not detectable by thin layer chromatography (TLC). The solvent was evaporated under vacuum after the reaction was complete. The crude precipitate was then dissolved in methanol and re-precipitated three times using ethyl acetate. The filter cake was dried at 30 °C for 12 h under vacuum until NB appeared as a light-yellow powder (0.4 g, 1.2 mmol, 66.7%). HA-NB was synthesized according to a published report[79]. Briefly, HA (408 mg, 1 mmol of disaccharide unit, Dongyuan Biotech, Zhenjiang) was dissolved in 50 mL deionized water at room temperature and NB (224 mg, 0.69 mmol) was added followed by HOBt (153 mg, 1 mmol, Sigma-Aldrich). The pH of the mixture was adjusted to pH 4.5, and the 1-(3-Dimethylaminopropyl)−3-ethylcarbodimide hydrochloride (EDC) (200 mg, 1.04, Sigma Aldrich) was added to the mixed solution and then stirred at room temperature for 48 h. Then, the solution was loaded into dialysis tubing (Molecular Weight (MW) cutoff 3500, Spectrum®) and dialyzed against diluted HCl (pH 3.5) containing 0.1 M NaCl for two days, then dialyzed against deionized water for 2 days. The solution was lyophilized and HA-NB was obtained in powder form.

## Synthesis of the photo-initiator

The photo-initiator lithium phenyl-2,4,6- trimethylbenzoylphosphinate (LAP) was synthesized based on our previous research[37,38]. In brief, Dimethyl phenylphosphonite (Ourchem) was reacted with 2,4,6-trimethylbenzoyl chloride (Sigma-Aldrich) via a Michaelis–Arbuzov

reaction. At room temperature and under argon gas, 3.2 g (0.018 mol) of 2,4,6-trimethylbenzoyl chloride was added dropwise to an equimolar amount of continuously stirred dimethyl phenylphosphonite (3.0 g). The reaction mixture was stirred for 18 h whereupon a four-fold excess of lithium bromide (Aladdin, 6.1 g) in 100 mL of 2-butanone (Sinopharm Chemical Reagent) was added to the reaction mixture from the previous step, which was then heated to 50 °C, a solid precipitate had formed after 10 min. Then, the mixture was cooled to ambient temperature and allowed to rest for 4 h, and then filtered. The filtrate was washed and filtered 3 times with 2-butanone to remove unreacted lithium bromide, and excess solvent was removed by vacuum.

## The preparation of the GelMA/HA-NB/LAP and GelMA/HA-NB/LAP/NSS hydrogels for 3D printing

For the preparation of GelMA/HA-NB/LAP and GelMA/HA-NB/neurotrophic supplements (NSs) hydrogels, the freeze-dried GelMA foams and HA-NB foams were dissolved in PBS solution at 40 °C. The photo-initiator LAP was added to the completely dissolved hydrogel solution. The final GelMA/HA-NB/LAP hydrogel solution was composed of 5% GelMA, 1.25% HA-NB and 0.1% LAP. For the precursors of GelMA/HA-NB/LAP/NSs hydrogel, the 2 X concentrated NSs was added to the GelMA/HA-NB/LAP hydrogel solution.

## The DLP-based 3D printing of scaffolds

The GelMA/HA-NB/LAP and GelMA/HA-NB/LAP/NSS hydrogel solution was prepared according to above methods and 0.04% phenol red was added to make a final bioink solution for further printing. A custom-made digital light-processing (DLP) 3D printer was utilized to fabricate the scaffolds. Specimens were designed by CAD software (AutoCAD 2019, Product Version P.45.M.377) and saved as a Stereo Lithography (STL) file. STL files were sliced in the Z direction and projected into the designed 3D morphology. Printing was carried out by repeated process of projecting an image into the bioink followed by raising the Z-stage. The printing process can be described briefly as follows. GelMA/HA-NB/LAP bioinks were added in the container of DLP printer and the designed scaffold was printed according to the STL file. After the GelMA/HA-NB/LAP scaffolds were finished, the container was washed three times by sterile PBS. Then, the GelMA/HA-NB/LAP/NSs bioinks were added in the washed container and the printing process was going on like before. For further cell and animal experiments, all these printing processes were carried out under aseptic conditions. Specially, every scaffold was washed by sterile and warm PBS to remove untreated solution. The GelMA/HA-NB/LAP scaffolds immersed in sterile PBS, and GelMA/HA-NB/LAP/NSs scaffolds were immersed in culture medium with neurotrophic supplements at a same concentration with that in bioink to avoid the diffusion. Printing parameters were used as follows: line width, 100 µm; pore size, 200 µm; printing thickness, 50µm; curing time, 2 s per layer; light intensity, 75 mV/cm². Cylindrical flake was printed with a 8 mm-diameter and 2 mm-thickness for rat experiments; or 3mm-diameter and 1mm-thickness for mice experiments.

## Live/dead staining assay

$1 \times 10^5$ cells were added to per GelMA/HA-NB/LAP or GelMA/HA-NB/LAP/NSs scaffolds to evaluate the cytotoxicity of hydrogels. And the cells were cultured on the scaffolds in DMEM media supplemented with 10% fetal bovine serum (FBS) and 1% penicillin/streptomycin at 37 °C and incubated at 5% CO₂ for 7, 14, and 21 days. Images were captured by a laser scanning confocal microscope (OLYMPUS IX83-FV3000, Japan) for live/dead assay.

## Animals and surgical procedures

SD rats (~ 250 g, 8-10 weeks old, Male) were used in this study. All surgical procedures were performed under isoflurane (4%) anesthesia.

Surgical sites were sterilized using a complex iodine solution after hair removal using a clipper. For subcutaneous transplantation model, an incision with the length of 1.5 cm was made in the mediodorsal skin and a lateral subcutaneous pocket of SD rats was prepared. 3D printed hydrogel scaffolds with or without NSs (cylindrical flake: 8 mm in diameter and 2 mm in thickness) were implanted under sterile conditions. At 1, 2, and 4 weeks, the rats were sacrificed and the samples were processed for histological analysis. For calvarial defect model, a critical-sized cranial defect with a diameter of 8 mm was created in the center of the calvarium using dental trephine. The rats were divided into three groups: (1) Defect control group (Defect), (2) Scaffold only group (Scaffold), (3) NSs-loaded scaffold group (Scaffold + NSs). At 1 week and 2 weeks after surgery, the rats were sacrificed and the regenerative tissues in the defect area were harvested and digested for single-cell collection. At 4 weeks and 12 weeks after surgery, the rats were sacrificed and the calvarium were harvested for further assessments.

All mice (12-15 weeks old, Male) were maintained in a C57BL/6 background. Prx1-Cre mice (JAX stock #005584) and Rosa26-loxp-stop-loxp-tdTomato mice (JAX stock #007905) were provided by Professor Weiguo Zou. Prx1-Cre mice crossed with Rosa26-Ai9 mice to specifically express tdTomato fluorescence in Prx1-expressing cells. All mice were bred and maintained under Specific Pathogen Free (SPF) conditions in the institutional animal facility of the Shanghai Institute of Biochemistry and Cell Biology, Chinese Academy of Sciences. For calvarial defect model, a cranial defect with a with a diameter of 3 mm and thickness of 1 mm was created in the left side of each calvarium using dental trephine. The mice were divided into two groups: (1) Defect control group (Defect), (2) NSs-loaded scaffold group (Scaffold + NSs). At 2 week and 4 weeks after surgery, the mice were sacrificed and the regenerative tissues in the defect area were harvested for further assessments.

The rats and mice used in this study were fed in separated cages in a temperature, humidity-controlled (-25 °C, 50–80%) and 12 h light/dark cycle room. All animals were treated according to standard guidelines approved by the Zhejiang University Ethics Committee (Ethical NO. ZJU20210064).

## Micro CT analysis and paraffin embedding

After the defect surgery for 4 and 12 weeks, rats ($n = 3$ rats, per group, per time point) were euthanatized and the calvarial specimens were harvested and fixed overnight with 4% paraformaldehyde at 4 °C. The fixed samples were scanned using micro-CT (U-CT-XUHR, MILabs) at 4 μm resolution. The three-dimensional (3D) structures of calvarium were reconstructed through MILabs-Rec interface, and analyzed by IMALYTICS Preclinical 2.1. software of the micro-CT. A cylinder space representing the region of interest (ROI) was designated to evaluate both bone and tissue volume for calculation of bone volume/tissue volume (BV/TV) and Bone density (BMD). Samples were then decalcified with 0.5 M EDTA for 8 weeks and then subjected to paraffin embedding.

## Histological analysis

The harvested specimens were fixed and decalcified in 10% ethylene-diamine tetra-acetic acid (EDTA) for two months and then dehydrated through graded alcohol series and embedded in paraffin. Sections of the central segment were cut into 10 μm thick slices using a rotary micro-tome (Leica, Hamburg, Germany). Hematoxylin and Eosin (H&E) staining, Safranin-O staining and Masson's Trichrome staining were performed on paraffin sections according to standard protocols, and observed using bright-field microscopy.

## Immunofluorescent staining

Cells cultured on dishes were fixed with 4% paraformaldehyde (PFA) for 20 min and then permeabilized with 0.03% Triton X-100 for 10 min at room temperature. After washing with PBS for 3 times, the samples were incubated with blocking solution (2% bovine serum albumin) for 30 min at room temperature to prevent nonspecific binding. The primary antibodies were diluted 200 or 500-fold with blocking solution and added to the cell cultures at 4 °C overnight, including Ki67 (1:250 dilution, ab16667, Abcam, USA) and γh2AX (1:250 dilution, ab22551, Abcam, USA). Tissues were fixed with 4% PFA and the paraffin-embedded samples were cut into 10-μm-thick sections. The tissue samples were deparaffinized and the antigens were activated by heating the slides in 10 mM citrate buffer (pH 6.0) at 65 °C overnight. After treating the sections with 0.3% H2O2 in MeOH, the samples were incubated with blocking solution (5% bovine serum albumin and 0.03% Triton X-100 in PBS). The samples were treated with primary antibodies at 4 °C overnight, including CD200 (1:100 dilution, AF2724, R&D Systems, USA), OCN (1:50 dilution, MAB1419, R&D Systems, USA), NGFR (1:50 dilution, NBP2-67296, NOVUS, USA), CD31 (1:100 dilution, ab222783, Abcam, UK), Msx1 (1:200 dilution, NBP2-57969, NOVUS, USA), Mmp13 (1:100 dilution, NBP2-45887, NOVUS, USA), Aggrecan (1:100 dilution, NB600-504, NOVUS, USA), RUNX2 (1:200 dilution, ab76956, Abcam, UK), COL2A1 (1:50 dilution, sc-52658, Santa Cruz, USA), SOX9 (5ug/ml, ab185966, Abcam, UK), Collagen X (1:100 dilution, 14-9771-182, Invitrogen, USA). After incubation with primary antibody, samples were then incubated with Alexa Fluor® secondary antibodies (G-Rabbit Alexa Fluor® 488, A11008; Goat anti mouse Alexa Fluor® 488, A11001; G-Rabbit Alexa Fluor® 546, A21430-f; G-Rabbit Alexa Fluor® 647, A32733; Invitrogen, USA and G-Mouse Alexa Fluor® 647, 4410 S, CST, USA) (diluted 1:500) for 1 h at room temperature. After incubation, the nuclei were stained with 0.1 μg/ml DAPI (Invitrogen, USA). After staining, the samples were observed using a confocal microscope (OLYMPUS IX83-FV1000 and FV3000-OSR, Japan).

## Immunohistochemistry staining

For immune-histochemical staining, sections were prepared, and followed by antigen retrieval by heating the slides in 10 mM citrate buffer (pH 6.0) at 65 °C overnight, inactivation of endogenous peroxidase by hydrogen peroxide with 0.3% H2O2 in MeOH, BSA blocking, and was incubated with primary antibodies against OCN (1:50 dilution, MAB1419, R&D Systems, USA), at 4 °C overnight. Then sections were incubated with anti-Rabbit secondary antibody conjugated with 1:1000 HRP (Beyotime Institute of Biotechnology). The stained specimens were photographed digitally and viewed under the Digital Slide Scanners.

## Single-cell RNA Sequencing and data analysis

At 1 week and 2 weeks after surgery, the rats (male, $n = 6$ biological rats, per group, per time point) were sacrificed. The regenerated tissues in calvarial defect area were minced with razor blades and washed several times by 4 °C PBS, and then digested by enzyme mixture (type I collagenase 0.1% and type II collagenase 0.1%, incubated at 37 °C for 40 min) before filtered through a 70 μm nylon mesh to obtain single cell suspension. Subsequently, the mRNA libraries were prepared (10x Genomics) and sequenced.

For quality control, cells with over 300 genes detected, 500 reads count and less than 20% of mitochondrial gene expression were kept. After filtering, we obtained data from 26016 cells: with 7749 cells in defect group, and 7712 cells in NSs-treated group of samples from 1 week after surgery; and 5420 cells in defect group, 5135 cells in NSs-treated group of samples from 2 weeks after surgery. Following the Seurat pipeline[39], the data was log-normalized, then scaled. Uniform Manifold Approximation and Projection (UMAP) and t-distributed Stochastic Neighbour Embedding (t-SNE) were calculated to visualize cell heterogeneity in reduced dimensions.

For single-cell clustering and annotation, after dimensionality reduction, we performed clustering using FindClusters function offered by Seurat. Genes expressed specifically in each clusters were

calculated by Seurat FindAllMarkers function. To identify the biological cell type of each cluster, we performed SingleR[80] analysis, combined with conventional markers of some known cell types. Firstly, 26,016 cells were clustered into major cell types and then further clustered into 33 subclusters totally. For differentiation trajectory analysis, to map differentiation in bone regeneration, we performed pseudotime analysis with R package Monocle[57] (version 2.20.0). Specifically, we computed trajectory of osteo-lineage cells and compared different states of cells using BEAM function provided by Monocle. RNA velocity was performed with scvelo python package (version 0.1.25). For gene functional annotation analysis, GO enrichment analysis was performed for markers of single-cell clusters using clusterProfiler[81] package. The enriched GO terms were filtered by setting pvalueCutoff to 0.01. For single-cell regulatory network analysis, the analysis of single-cell gene regulatory network was performed using the SCENIC[82] package followed by the standard pipeline. Dot plot shows the cell-type specific regulons with top Regulon Specificity Score (RSS)[83] and their average expression (Z) in the cell subtype. For cell-cell interaction analysis, interaction among cells was analyzed using R package iTALK[84] following the manual provided by the package. Differential analysis was made between NSs group and control using Wilcoxon rank sum test.

## Statistical analysis

Values are expressed as mean ± SD or mean ± SEM unless otherwise indicated in the figure legends. The significance between two groups was analyzed using two-tailed Student's $t$-tests. For multiple comparisons, one-way analysis of variance (ANOVA) with Tukey's post hoc test was used. Statistical analysis was performed using the Graphpad software. $P < 0.05$ was considered to be significant; *$P < 0.05$, **$P < 0.01$, ***$P < 0.001$, ****$P < 0.0001$.

## Reporting summary

Further information on research design is available in the Nature Research Reporting Summary linked to this article.

# Data availability

The single cell RNA-sequencing data generated in this study have been deposited in the Genome Sequence Archive (GSA) database with accession number CRA005302, which is now publicly available can be accessed from the following "link [https://bigd.big.ac.cn/gsa/browse/CRA005302]". All other relevant data supporting the key findings of this study are available within the article and its Supplementary Information files or from the corresponding author upon reasonable request. Source data are provided with this paper.

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

## Acknowledgements

This work was supported by the National Key Research and Development Program of China (2017YFA0104900), and NSFC grants (NO. T2121004, 31830029). The authors thank Weiliang Shen (The Second Affiliated Hospital of Zhejiang University) for providing the human bone marrow samples from trauma subjects in traffic accidents or discarded tissue during the operation of fractures. The patients or their relatives' consent, as well as approval of the local ethics committee were obtained prior to harvesting of human tissue samples. We thank Wei Yin and Lijun Xuan (Core Facilities, Zhejiang University School of Medicine) for their assistance with Confocal laser scanning microscope.

## Author contributions

Xianzhu Zhang and Hongwei Ouyang designed the project, performed experiments and wrote the whole manuscript; Xianzhu Zhang, Wei Jiang, Yishan Chen and Shufang Zhang helped revised the manuscript; Xianzhu Zhang, Chang Xie and Xinyu Wu performed the in vitro cell culture and osteogenic induction assays with optimized neurotrophic supplements; Xianzhu Zhang and Chang Xie completed the hydrogel scaffold preparation, animal experiments and the histological analysis; Chang Xie and Yuqing Gu helped with the immunofluorescence staining of regenerated tissue sections; Xinyu Wu, Fei Wang and Yi Zhang performed flow cytometry analysis; Xianzhu Zhang, Wei Jiang and Xilin Shen performed single-cell sequencing and data analysis; Hongwei Wu, Youguo Liao, Renjie Liang, Wei Sun and Tao Zhang helped with the animal experiments; Qian Ren and Prof. Weiguo Zou helped with the breeding of Prx1-Cre; Rosa26-Ai9 mice and lineage-tracing experiments; Yi Hong and Wei Wei helped with the material preparation and DLP printing.

## Competing interests

The authors declare no competing interests.
