## [Peer Review File · Nature Communications]

REVIEWER COMMENTS

Reviewer #1 (Remarks to the Author):

The paper presents a very complete and detailed work on the potential of neurotrophic factors release from a 3D printed hydrogel to recruit specific regenerative cells in a skull defect.

The English language requires a complete (preferably professional) revision.

There are however major issues that the authors should address more appropriately before acceptance of the paper, namely:

- 1) More information and details on the cocktail of the neurotrophic factors used and their concentrations, how these relate or not specifically to the subset of cells that were preferentially recruited?
- 2) How relevant is to use 3D printing for this particular approach (instead of instances of injecting the hydrogel with the NSs)?
- 3) More detail on the properties of the gel itself, how does it compare to the large amount of similar materials that have been proposed for the same applications?

Reviewer #2 (Remarks to the Author):

This paper proposes that neurotrophic supplements improve the bone-forming capability of human bone marrow cells, and therefore may add value to stem cell therapies for the treatment of large-bone defects.

I am afraid the paper has several significant issues that have to be carefully addressed. First, the entire manuscript is full of grammar errors and incomplete sentences that distract the reader and make it challenging to focus on the actual story. This reads more like a 1st draft than a carefully proof-read manuscript. Further, the figures are stuffed with panels and legends with very small fonts, which makes it difficult to follow the narrative.

The methods and results are also presented in an unclear manner. As a result, I cannot find the findings of this study compelling. For example, the BMSC collection methods (which are at the core of everything that has been done) are written in a confusing way: It sounds like you first collected matrix-free marrow

cells, and then you digested them with collagenase— why? It also sounds like you merely passaged these cells in vitro, and then assumed that what you obtained are stem cells. Were any other sorting or purification steps involved? Some cells are identified as stem cells merely based on CD200 and ITGAV transcripts; I don't think this alone qualifies them as stem cells.

Following the previous point: It seems that you xenografted cultured human marrow cells into a rat calvarial defect. Did you distinguish between human cells and rat cells that may have migrated into the defect site during the repair process? Further, if you implanted merely MSCs, how come the majority of the cells you detect with single cell RNA-seq are CD45+?

The nomenclature for genes and proteins used by the authors add to the confusion I mentioned above. Do you refer to human or rat genes with non-capitalized gene symbols (e.g. Col1a1 in Figure 3d)? For immunohistochemistry, have you validated the specificity of your antibodies such that you can distinguish between rat and human cells?

The changes in cellular diversity you highlight in Figure 4 should be validated with FACS, as it is unclear how many replicates the single cell RNA-seq data are based on, and how robust these changes are. The immunohistochemistry images in the small panels are difficult to interpret, as the legends are tiny and the contrast is low for some images, such as Msx1.

An important point the authors seemingly disregard is that abundance or pseudo-time analysis on a single cell RNA-seq dataset does not prove that a certain cell population has lineage relationship with others. This can only be proven with classic lineage-tracing strategies, wherein the population of interest is permanently marked with a reporter protein.

Some minor points:

Introduction is too long – at 3 single-spaced pages. A detailed review of what other groups have done individually (such as the references to work by Li and Clemens labs) is not necessary – these should be mentioned concisely within the context of your work.

Line 224 – supp fig2: what do the yellow arrowheads indicate?

You talk about subcutaneous implantation all of a sudden (line 228) with no details? What animal, under what circumstances?

Please provide more details on the critical sized defect. It sounds like this was centered on the sagittal suture, extending into the parietal bones. Were any of the other sutures damaged? A schematic would help.

Figure 2: what does the dashed line indicate in Fig2o?

There is no mention of NGFR in the text or figure caption, what does NGFR immunostaining indicate?

Some clusters are labeled osteoblasts, however it is unclear whether they express the mature bone-forming cell marker Bglap.

Reviewer #3 (Remarks to the Author):

The authors address the single-cell RNA analysis of the cells recruited to their in situ culture system, enriched with neurotrophic supplements (NSs). Besides mapping the cellular subpopulations in a cranial defect model in rats, they describe its bone regenerative capacity. Main findings include endochondral regeneration and an MSX1+ population associated with it.

Major concerns:

Though the study is very extensive a few major concerns should be addressed:

1. The authors describe that intramembranous ossification is the developmental origin of cranial tissue which is well known indeed. However, craniomaxillofacial repair may also proceed through endochondral ossification and should not be excluded as a natural phenomenon as such.
2. Overall the introduction is long and at times not relevant to the aim of this study. The focus of the paper is not completely clear based on the introduction since it starts with bone defect regeneration while the data presented here in that respect are not ground breaking. I would have expected a larger focus on the roles and identification of specific SSC subpopulations which is where the novelty of this study may lie but only if point 1 can be sufficiently motivated.
3. In general, I do not see sufficient novelty in the presented findings with a large focus on bone regeneration (not sufficient novelty).
4. The complete study is built around the use of NSs as attractor of the stem cells and stimulator of regeneration. The rationale for the use of this specific cocktail is missing. The introduction only describes the use of soluble factors as a means to induce regeneration but fails to explain why NSs were used.

Why devote so much effort (single cell analysis) to this specific formulation? The regenerative outcome was not even complete regeneration of the defect site, so how is this particular cocktail of more interest than comparably regenerative formulations? Also, at times optimized composition of the cocktail was indicated while this was not part of this study, nor optimized for in vivo application in cranial defects as used here. Finally, the composition of the NSs is disputable from the perspective that FBS is included which is known to have a large batch-to-batch variation that can have a major impact on any study in bone regeneration.

5. Description of results indicating significant differences are not always supported by actual quantitative data (Ki67, gammaH2AX, cell expansion/cell numbers, proportion of living cells) and therefore the associated conclusions regarding the local expansion of cellular subpopulations cannot be truly supported. Also, 2D quantification of chondrogenic differentiation is not very representative (3D cultures should be shown with preferably GAG but also collagen type II identification).

6. The cell expansion conclusion is not sufficiently supported by the data. FGF which is in the NSs for example, is known to increase cellular proliferation but also known to stimulate subsequent differentiation of cells. But only showing qualitative data on larger numbers of MSX1+ cells over time does not confirm targeted cell expansion of this subpopulation. Was cell expansion not stimulated for all cells/subpopulations present or selectively for the MSX1+ population? And were the MSX1+ cells actually proliferating or acquiring the MSX1+ phenotype after proliferation. These data are not supplied and therefore the conclusions on cell expansion of a specific subtype not supported.

7. Increased osteogenic capacity (ALP, ARS) were not normalized to cell numbers. Since cell expansion may be affected differently under the various conditions, this correction on the data is required.

8. Why were microchannels of 200um generated in the printed constructs? It is generally known that around 400um is optimal for bone regeneration purposes.

9. I may have missed it but I would have liked to see a discussion of the presence of MSX1 on these cells. MSX1 is associated specifically with dental development and not typically associated with bone regeneration in the skeleton outside of the head region. And if it's a chondro-repressor, how can it be co-expressed with col6 and aggrecan? Also, endochondral bone regeneration could be confirmed by staining for markers such as collagen type X.

10. At some point a femur defect was introduced which was a bit confusing (Suppl fig 13). This was not described in the methods. Here local cells from the mesenchymal rather than neural crest origin will be recruited with a different differentiation capacity. Why were these data added and what are they showing? Were the recruited cells also MSX1+ here? What is the point fo showing these data?

11. I would suggest rewriting of the manuscript by a native speaker since many errors are present, some of which hinder proper understanding of the manuscript.

Point-to-point response

Reviewer #1 (Remarks to the Author):

The paper presents a very complete and detailed work on the potential of neurotrophic factors release from a 3D printed hydrogel to recruit specific regenerative cells in a skull defect.

The English language requires a complete (preferably professional) revision.

Re: Thanks for your patient work in reviewing this paper, we have now thoroughly revised the manuscript and then improved the English language by the Springer Nature Author Services.

There are however major issues that the authors should address the more appropriately before acceptance of the paper, namely:

1) More information and details on the cocktail of the neurotrophic factors used and their concentrations, how these relate or not specifically to the subset of cells that were preferentially recruited?

Re: Thanks for the reminding of detailed information regarding the cocktail of neurotrophic factors.

a) We have revised and provided the information on the type and concentrations of the NSs ingredients used in this research in the "Materials and Methods" section.

b) Previous studies have shown that neurotrophic signaling is required for the osteochondral progenitor expansion during bone development [1] and cranial bone injury repair [2]. In the present study, the SSC2 subset was marked by the specific expression of Msx1, which is a transcription factor that played an important role in the craniofacial bone morphogenesis [3]. Together, these findings indicate a high correlation between the Msx1⁺ SSC2 subset and the cocktail of the neurotrophic supplements (NSs).

c) In our data, we found the Msx1⁺ SSC subset namely Skeletal Stem Cells 2 (SSC2) was preferentially recruited by the NSs-contained in situ culture for 2 weeks. To further understand the mechanism, the transcriptional signature of the SSC2 subset itself revealed by our single-cell RNA-Seq data could partially help explain the cellular response to the neurotrophic supplements (NSs). Main components in the NSs including FGF2, EGF, StemPro® Neural Supplement, and B27 Supplement, were previously known for promoting cell migration, proliferation, or cell differentiation. After in situ transplantation of NSs-loaded gel scaffold for two weeks, GO analysis of DEGs in SSC2 shows much stronger ability in cell migration, proliferation and differentiation. (as below).

GO terms of DEGs in SSC2 showing featured functions of SSC2 after treatment with NSs.

Dot size indicates gene ratio and color represents adjusted p value.

d) Besides, specific factors of neurotrophic supplements, such as FGF2 and StemPro® Neural Supplement (Gibco), two of the main components in NSs, are previously used for the culture of neural stem cells (NSC). On one hand, SSC2 is similar to NSCs to some extent. Correlation analysis between the four skeletal stem/progenitor subsets in our data and an external NSC cohort (GSE100320) [4] showed that SSCs have much higher correlation with NSC ($r \geq 0.5$) than other cell subsets, indicating the similarity in gene expression pattern with NSC (as below). On the other hand, SSC2 express significantly much higher *Fgfr2*, receptor of FGF2, among the four skeletal stem/progenitor cell subsets, indicating a more specific stimulation of SSC2 by NSs (as below).

Correlation between skeletal stem/progenitor cells and neural stem cells in other study.

Color indicates the correlation score (pearson) between cells.

Expression of Fgfr2 in four skeletal stem/progenitor cells.

2) How relevant is to use 3D printing for this particular approach (instead for instances of injecting the hydrogel with the NSs)?

Re: Thanks for the comment.

a) For the critical-sized bone defects, a scaffold with defined porous structure and mechanical strength was preferred than hydrogel injection to provide more adhering surfaces and mechanical support for endogenous stem cells and blood vessel cells growth and migration during the bone repair process [5].

b) Besides, a form of 3D-printed hydrogel scaffold enables more efficient oxygen and nutrient exchange, and is beneficial for the necessary vascularization, osteogenic differentiation, and the subsequent calcium deposition [6].

3) more detail on the properties of the gel itself, how does it compares to the large amount of similar materials that have been proposed for the same applications?

Re: Thanks for the comment.

a) The multiple properties of the gel itself: Our previous study [7] has developed the photo-reactive adhesive (GelMA/HA-NB/LAP gel system) that mimics the extracellular matrix composition as an tissue-adhesive hydrogel for the rapid seal of bleeding after UV light irradiation. The detailed chemical structure, schematic diagram of the photo-triggered imine-crosslinked matrix hydrogel, dynamic time-sweep rheological analysis, and torsion modulus, as well as the gel points were presented below.

Chemical structure and mechanical properties of the hydrogels. a. Constituent chemical structures and a schematic diagram illustrating the formation of the photo-triggered imine-crosslinked matrix hydrogel. b. To monitor the gelling process, a dynamic time-sweep rheological analysis was carried out with an in situ photo-rheometer (HAAKE Mars III, light, Omnicure S2000 365 nm: 30 mW cm⁻²) showing the formation kinetics for GelMA/HA-NB/LAP, GelMA/HA-NB, and GelMA/LAP hydrogels. c. The final torsion modulus G' of different hydrogels. d. The gel point of different hydrogels. All the gelling measurements were conducted using OmniCure S2000 (365 nm, 30 mW/cm²). Exposure time: 180 s for GelMA/HA-NB/LAP and GelMA hydrogels, and 300s for GelMA/HA-NB hydrogel.

b) Other similar materials that have been proposed for the same applications: Dopamine (DA)-modified alginate hydrogel incorporated with HAp microparticles (MPs) and stem cell aggregates [8], which is a visible-light crosslinkable adhesive hydrogel, was used as an injectable MSC delivery vehicle for bone tissue engineering applications. And the bone regeneration was largely due to the presence of MSC aggregates and HAP microparticles.

Microporous networked photo-crosslinkable chitosan hydrogel incorporated with nanoclay particles [9] with increased mechanical properties, which is also a visible-light crosslinkable hydrogel, to promote native cell proliferation and attachment for direct bone formation.

c) Unique properties of the gel itself: On one hand, the GelMA/HA-NB/LAP gel system used in our study is a UV-light crosslinked adhesive hydrogel with a tough mechanical property, which means this was almost the first time of UV-light crosslinked hydrogel that applied in the bone tissue engineering. On the other hand, instead of a direct injection, the 3D printed GelMA/HA-NB/LAP gel scaffold containing NSs medium provided as a unique in situ cell culture condition for endogenous stem cells. Therefore, the GelMA/HA-NB/LAP gel scaffold without HAp incorporation in this study functions both as a drug delivery vehicle and a pro-regenerative niche.

d) The advantages of the GelMA/HA-NB/LAP gel compared with other similar materials that used for the bone repair applications: Our research developed a biomimetic bioink composed of GelMA, HA-NB, and the polymerization initiator LAP, which are suitable for DLP-based printing. The GelMA/HA-NB/LAP gel was chosen for its fast gelation properties and its structural similarity to the natural ECM [7]. Besides, the printed scaffold holds microporous structures, and is beneficial for neovascularization and endogenous cell migration [10]. During bone regeneration process, multiple cell types including mesenchymal stem cells, endothelial cells, nerve cells and macrophages are recruited to the defect sites [11]. The main ingredient GelMA has been proved to be highly biocompatible to all of those cells [12], showing its potential for bone repair. Moreover, compared with synthetic polymers and bioceramics, GelMA has been revealed in transcriptomic analysis to promote bone regeneration and reconstruction by improving blood vessel formation through enhancing oxygen transportation and red blood cell development [12]. Combining HA-NB, the mechanical properties of the printed scaffold can be easily tuned as the UV-light induces the aldehyde group to react with the amine group on the GelMA chain [7]. Lastly, the GelMA /HANB/LAP hydrogel may better help the osteointegration, as it can bond to the surrounding tissues tightly through Schiff based reactions [10].

Reviewer #2 (Remarks to the Author):

This paper proposes that neurotrophic supplements improve the bone forming capability of human bone marrow cells, and therefore may add value to stem cell therapies for the treatment of large-bone defects.

I am afraid the paper has several significant issues that have to be carefully addressed. First, the entire manuscript is full of grammar errors and incomplete sentences that distracts the reader and makes it challenging to focus on the actual story. This reads more like a 1st draft than a carefully proof-read manuscript. Further, the figures are stuffed with panels and legends with very small fonts, which makes it difficult to follow the narrative.

Re: We appreciated your patience and wise suggestions for reviewing this manuscript, we

have now thoroughly revised the manuscript and the main figures, as well as improved the English language by the Springer Nature Author Services.

The methods and results are also presented in an unclear manner. As a result, I cannot find the findings of this study compelling. For example, the BMSC collection methods (which are at the core everything that has been done) are written in a confusing way: It sounds like you first collected matrix-free marrow cells, and then you digested them with collagenase– why? It also sounds like you merely passaged these cells in vitro, and then assumed that what you obtained are stem cells. Were any other sorting or purification steps involved? Some cells are identified as stem cells merely based on CD200 and ITGAV transcripts; I don't think this alone qualifies them as stem cells.

Re: Thanks for the comments and kind reminding. The detailed progress of BMSC isolation was described by mistake, we have now revised the corresponding information in the “Materials and Methods” section. If we understand correctly, there are two distinct aspects regarding the above questions, and some of which can be explained as follows:

a) In this study, human BMSCs were in fact isolated and cultured according to published protocols in the literatures [13, 14], namely, BMSCs were isolated using a Percoll gradient (Gibco) and cultured in low-glucose Dulbecco's modified Eagle's medium (DMEM; Gibco) containing 10% fetal bovine serum (FBS; Gibco), 1% penicillin/streptomycin (P/S; Thermo Fisher Scientific), and incubated at 37°C, 5% CO₂. For the verification of the human MSC identity before use, according to the basic definition of mesenchymal stem cells, flow cytometric analysis was performed by us to confirm that these cells were positive for MSC surface markers CD73, CD90, and CD105, and negative for hematopoietic cell markers CD34 and CD45. Passage 3-5 of the BMSCs were then utilized for the *in vitro* experiments in this study.

Flow cytometry analysis of mesenchymal stem cell markers

b) For the evaluation of NSs-loaded scaffold in rat calvarial defect, cells from the regenerated tissues within defect area were digested with collagenase and isolated for single-cell RNA sequencing and analysis. The detailed information about the tissue digestion and collection of single cell suspension can be found in the “Single-cell RNA Sequencing and data analysis” part of the “Materials and Methods” section.

Previous studies that done by Michael T. Longaker group [15] have revealed that [CD45⁻Ter-119⁻Tie2⁻AlphaV⁺Thy6C3⁻CD105⁺CD200⁺] cell populations isolated from femora of mice and identified as skeletal stem cells. Even lacking Thy1 and 6C3 expression, periosteal mesenchymal stem cells (PSCs) were identified by using CD200 and CD105 (CD200⁺CD105⁻ periosteal mesenchymal stem cells (PSCs), CD200⁻CD105⁻ periosteal

progenitor 1 (PP1) cells, and CD105⁺CD200^{variable} periosteal progenitor 2 (PP2) cells among CTSK-mGFP⁺ population by Matthew B. Greenblatt group [16]. Regarding the identity of the two SSCs (SSC1 and SSC2) subsets that we distinguished in the single-cell RNA sequencing data analysis, stem cell markers CD90 (Thy1), CD105 (Eng), CD29 (Itgb1) and Pdgfra were found among these cell subsets (figures are as below). Besides, the “skeletal stem cell subsets” in our data expressed both CD200 and Itga5. Together with their upstream role in the differentiation hierarchy (Figure 4), SSC1 and SSC2 in our data were identified as skeletal stem cells.

Violin plot shows the expression levels of stem/progenitor cell markers among the osteo-lineage cell population.

Moreover, we performed an integrated analysis on our single-cell data and external data set (GSE108892) of stem/progenitor cells in bone marrow microenvironment [17]. UMAP plot shows a great overlap between our stem/progenitor cells (SSC2 subpopulation) and the Lepr⁺ stem/progenitor cells, validating our definition of stem/progenitor cells. Results are shown as follows.

UMAP plot showing the distribution of osteo-lineage cell population from our single-cell data and the external data (GSE108892). Left panel, distribution of cells from different sample origins; right panel, distribution of cells of different subpopulations and shows the overlap between the

SSC2 subpopulation (purple) and the Lepr⁺ stem/progenitor cells (yellow).

Following the previous point: It seems that you xenografted cultured human marrow cells into a rat calvarial defect. Did you distinguish between human cells and rat cells that may have migrated into the defect site during the repair process? Further, if you implanted merely MSCs, how come the majority of the cells you detect with single cell RNA-seq are CD45⁺?

Re: Thanks for the comment.

a) In the present study, we actually transplanted the “in situ culture system”, which is pure hydrogel scaffold incorporating the neurotrophic supplements (NSs), not including human marrow cells, into a rat calvarial defect. The use of human BMSCs during the in vitro experiment is to test the efficacy of NSs or NSs-loaded scaffold in expanding mesenchymal stem cells.

b) As we have mentioned above, the cells detected with single-cell RNA-seq were all migrated or recruited from the rat calvarial tissues. Those CD45⁺ cells indicating that there are immune-lineage cell populations during the bone injury repair process.

The nomenclature for genes and proteins used by the authors add to the confusion I mentioned above. Do you refer to human or rat genes with non-capitalized gene symbols (e.g. Col1a1 in Figure 3d)? For immunohistochemistry, have you validated the specificity of your antibodies such that you can distinguish between rat and human cells?

Re: Thanks for the comment. We are so sorry about the confusion caused by our inadequate writing and expression. Similar to the previous question, cells detected in the regenerated tissues were all from the rat itself, therefore, the nomenclatures for genes and proteins are both referred to rat species.

The changes in cellular diversity you highlight in Figure 4 should be validated with FACS, as it is unclear how many replicates the single cell RNA-seq data are based on, and how robust these changes are. The immunohistochemistry images in the small panels are difficult to interpret, as the legends are tiny and the contrast is low for some images, such as Msx1.

Re: Thanks for the wise suggestion.

a) In Figure 4, the Osteo-lineage cells in our scRNA-seq data including 8 cell subsets, there was an obvious increase of Pdgfra⁺ progenitors and SSC2 subpopulations in cell proportion after 1 week and 2 weeks of bone defect, respectively. However, the relative cell proportion of Pdgfra⁺ progenitors was diminished to a negligible level after the second week post injury. In this study, the changes in cellular diversity are mainly reflected on the SSC2 cell proportion increase (nearly 60% fraction of the total osteo-lineage cells is SSC2) at 2 weeks after injury (Figure 3), thus we focused on the SSC2 subpopulation in the following study.

b) To validate the increase of SSC2 subpopulation by Flow Cytometry during the in vivo bone repair process, we did repeated the NSs-Scaffold transplantation with animal experiment again using the same rat calvarial defect model. After 2 weeks post-surgery, we collected and digested the regenerative tissue from NSs group and Defect group (n=4

rats) for single cell suspension, respectively. Interestingly, CD200, as a cell surface marker of mouse skeletal stem cells [15], is specifically expressed in the SSC2 subpopulation in our data. Through flow-cytometric analysis of markers against Msx1 and CD200, we found that there are only a mild increase of Msx1⁺ CD200⁺ cells after NSs delivery, which is shown below:

Flow-cytometric analysis of Msx1 and CD200 of regenerated cells in rat calvarial bone defects

c) Due to the extremely high proportion of immune-lineage cells and endothelial cells at the early regenerative stage, take 2 weeks after injury for example, the osteo-lineage cells are relative at a low proportion among the whole cell populations in regenerative tissue (Figure 3). We think that's why the ratio of CD200⁺ Msx1⁺ cells of the regenerated tissue in vivo at rat calvarial bone defects.

d) Next, to further confirm the cell proportion of SSC2 increase after the treatment of neurotrophic supplements (NSs), we performed the flow-cytometric analysis of CD200⁺ Msx1⁺ double positive cells of bone marrow-derived mesenchymal stem cells (BMSCs, P3) expanded by using the culture system of NSs-loaded 3D printed gel scaffold in vitro. As expected, we observed that the CD200⁺ Msx1⁺ cells were increased to 13.1% after 2 weeks of NSs-loaded scaffold culture, when compared to the control medium-loaded scaffold culture.

Flow-cytometric analysis of markers against Msx1 and CD200 of in vitro cultured BMSCs using our NSs-loaded 3D printed gel scaffold system

e) For scRNA-Seq analysis, regenerated tissues from 6 rats per group at each time point were provided, which we have updated in the “Materials and Methods” section. We are sorry for the tiny and low contrast images, and we now have revised these figures as mentioned above.

An important point the authors seemingly disregard is that abundance or pseudo-time analysis on a single cell RNA-seq dataset does not prove that a certain cell population has lineage relationship with others. This can only be proven with classic lineage-tracing strategies, wherein the population of interest is permanently marked with a reporter protein.

Re: Thanks for your professional advice and comments.

a) We were also aware of the importance of verifying the lineage relationship we have acquired from the single cell RNA-seq dataset. Firstly, we contacted research groups whom are in the developmental biology area and wish for a direct lineage-tracing mice and tracing for Msx1-marked SSC2 population in our study. As the domestic source of Msx1-cre; Rosa-tdTomato mice is extremely limited, we have to make an alternative plan to seek any other possibilities to prove the relationship.

b) Based on the scRNA-seq data, we found that *Prrx1*, which is a marker for the limb bud mesenchyme [18] and a upstream SSC marker [19], was almost expressed in all the osteo-lineage cell and perivascular cell clusters. While the *Msx1* expression was restricted in the osteo-lineage cell clusters (as shown below). We thus hypothesized that *Msx1*⁺ SSC2 would be a downstream progeny of *Prrx1*⁺ skeletal stem cells, together with lineage-specific marker staining, we could prove the “SSC2 – Osteoblasts” differentiation trajectory indicated in Figure 4.

Prrx1 expression among the all the subpopulations in our scRNA-seq data

Msx1 expression among the all the subpopulations in our scRNA-seq data

c) After transplantation of the NSs-loaded hydrogel scaffold in a calvarial defect model in the *Prx1*-cre; *Rosa*-tdTomato mice, the regenerated tissue samples were harvested for the analysis of *Prx1*-lineage traced *Msx1* positive cells and their progenies. Result are supplemented in Figure 7 in manuscript and also are shown as follows:

Msx1⁺ SSCs subset is a source of osteogenesis and closely correlates with endochondral ossification. **a.** Schematic of the assessment of Msx1⁺ SSCs subset in differentiation hierarchy by Prx1-lineage tracing mice. **b, c.** HE staining of newly formed calvarium tissues in skull bone injury at 2 weeks and 4 weeks after surgery, respectively. **d-g.** Coronal sections of the calvarium tissues from Prx1^{Cre}; Rosa^{tdTomato} mice co-immunostained with Msx1 and Runx2 (**d**), Msx1 and OCN (**e**) at 2 weeks; with Msx1 and ACAN (**f**), Msx1 and COL2 (**g**) at 4 weeks.

Some minor points:

Introduction is too long – at 3 single-spaced pages. A detailed review of what other groups have done individually (such as the references to work by Li and Clemens labs) is not necessary – these should be mentioned concisely within the context of your work.

Re: Thanks for the suggestion. We have revised the Introduction part accordingly and integrated these detailed review into the main context.

Line 224 – supp fig2: what do the yellow arrowheads indicate?

Re: Sorry for the unclear arrowheads. The yellow arrowheads in supp fig2d indicate red blood cells, and that in supp fig2e indicate endothelial cells. The revised text marked in yellow color was shown in the Supplementary Information.

You talk about subcutaneous implantation all of a sudden (line 228) with no details? What animal, under what circumstances?

Re: Thanks for the comment. To evaluate the efficacy of the local delivery of NSs in promoting bone regeneration, we first tested the effects of in vivo cell compatibility and vascularization of the hydrogel culture system with or without NSs by subcutaneous implantation before the in situ calvarial defects. After subcutaneous implantation at the dorsal site of SD rats, we found that the scaffold was gradually degraded with in four weeks, and NSs incorporation accelerated this progression as shown by obviously reduced gel area. We have now provided the corresponding details in “Animals and surgical procedures” of the “Materials and Methods” section.

Please provide more details on the critical sized defect. It sounds like this was centered on the sagittal suture, extending into the parietal bones. Were any of the other sutures damaged? A schematic would help.

Re: Thanks for your wise advice. For calvarial defect model, a critical-sized cranial defect with a diameter of 8 mm was created in the center of the calvarium using dental trephine, no other sutures are damaged. We have drawn a schematic to help understand the process and details of the animal experiment, which is shown in Figure 2 and as below:

The schematic of the rat calvarial defect model

Figure 2: what does the dashed line indicate in Fig2o?

There is no mention of NGFR in the text or figure caption, what does NGFR immunostaining indicate?

Re: Thanks for the comments and reminding.

a) The dashed line in the Fig2o indicates the border between the newly formed bone-like tissues. The description is now provided in the figure legend.

b) While sensory nerve fibers are required for the osteochondral progenitor expansion during mammalian skeletal development [20], NGF is the primary neurotrophin that stimulates skeletal sensory nerve ingrowth. Moreover, during bone formation, vasculature and innervation form prior to the onset of ossification. NGF-TrkA signaling also play an essential role of in bone healing [2, 20], and coordinates innervation, vascularization, and ossification [21]. In other studies, the level of innervation of in vivo implanted graft was an important aspect of for evaluation [22]. Therefore, NGFR immunostaining in our results are indicating the innervation level in bone healing, which we have added into the main text.

Some clusters are labeled osteoblasts, however it is unclear whether they express the mature bone-forming cell marker Bglap

Re: Thanks for the comment. Although we did not observe the expression of Bglap, this can be explained as follows:

a) On one hand, the regenerated tissues we collected for scRNA-seq analysis are newly formed soft tissues, in which none of the surrounding bone tissues were harvested and digested.

b) On the other hand, as 2 weeks after injury is relatively an early stage of bone healing, the maturation of newly formed osteoblasts may still take time. From our scRNA-seq data, we found the Runx2 and Alpl are co-expressed in our labeled osteoblasts, which indicate that an immature stage of these osteoblasts.

Violin plots showing specific Alpl gene expression levels in eight osteo-lineage population cells

Reviewer #3 (Remarks to the Author):

The authors address the single-cell RNA analysis of the cells recruited to their in situ culture system, enriched with neurotrophic supplements (NSs). Besides mapping the cellular subpopulations in a cranial defect model in rats, they describe its bone regenerative capacity. Main findings include endochondral regeneration and an MSX1+ population associated with it.

Major concerns:

Though the study is very extensive a few major concerns should be addressed:

1. The authors describe that intramembranous ossification is the developmental origin of cranial tissue which is well known indeed. However, craniomaxillofacial repair may also proceed through endochondral ossification and should not be excluded as a natural phenomenon as such.

Re: Thanks! We agree with your professional opinion and advice. To be more accurate, the craniofacial bones are formed mainly through intramembranous ossification [23]. Even though, the calvarial bone plates formation is mediated by intramembranous ossification [16, 24]. Therefore, our description is restricted on calvarial bone, not all craniofacial tissue repair.

2. Overall the introduction is long and at times not relevant to the aim of this study. The focus of the paper is not completely clear based on the introduction since it starts with bone defect regeneration while the data presented here in that respect are not ground breaking. I would have expected a larger focus on the roles and identification of specific SSC subpopulations which is where the novelty of this study may lie but only if point 1 can be sufficiently motivated.

Re: Thanks for the comment. We have revised the introduction section to be more concise and focused on the strategy of in situ delivery of soluble factors by recruiting specific skeletal stem cell subpopulations for more efficient bone repair.

3. In general, I do not see sufficient novelty in the presented findings with a large focus on bone regeneration (not sufficient novelty).

Re: Thanks for the comment. We think the novelty of this study for bone regeneration may largely lie on the following aspects:

a) Full-thickness regeneration of critical-sized calvarial defects was successfully achieved by implantation of a culture system of 3D printed hydrogel scaffold incorporated with a cocktail of neurotrophic supplements (NSs).

b) Single-cell RNA Seq analysis showed that implantation of the culture system greatly changed the atlas of in situ stem/progenitor cells and enhanced the local expansion of a specific Msx1+ SSCs subset, which was mainly responsible for the rapid bone formation.

c) The enhanced calvarial bone regeneration at the present study was accompanied with an endochondral ossification that closely correlated to Msx1+ SSCs.

4. The complete study is built around the use of NSs as attracter of the stem cells and stimulator of regeneration. The rationale for the use of this specific cocktail is missing. The introduction only describes the use of soluble factors as a means to induce regeneration but fails to explain why NSs were used.

Re: Thanks for the comment.

a) Previous studies have found that neurotrophins and neurotrophic tyrosine kinase receptor type A (NGF-TrkA) signaling directs innervations in bone tissues, and are required for osteochondral progenitor expansion during both bone development[25] and stress fracture repair[26]. Besides, the neurotrophins were also shown in cranial bone injuries, which in turn positively regulate bone repair through inducing innervation[27]. These studies together provided an available strategy and a neurotrophic mechanism of locally expanding osteogenic stem cell subpopulations for more efficient bone repair.

b) Other studies by using serum-free medium that containing neurotrophic supplements (NSs), which include N2, B27, epidermal growth factor (EGF), fibroblast growth factor (FGF)-basic, insulin-like growth factor-1 (IGF-1), platelet-derived growth factor (PDGF) and Oncostatin M (OSM), MSCs form mesensphere with higher self-renewing ability[28] and exhibit non-adherent spheroidization characteristics with improved microcirculation property[29]. In addition, MSCs showed a dynamic proportion of nestin+ staining in varied bone marrow area and their long-term cultures after the mesensphere culture with NSs-contained medium. Thus, we hypothesized that culture medium containing neurotrophic supplements (NSs) could improve the expansion of skeletal stem cell subpopulations.

c) From our previous results (which are not shown), we found that the proliferation of bone marrow mesenchymal stem cells (BMSCs) was significantly increased by using neural stem cells (NSCs) conditioned medium compared to the regular L-DMEM complete medium. Subsequently, we found a more stronger BMSCs proliferation when directly adding the neurotrophic factors to the culture medium. Some of the results are shown as below:

Proliferation of bone marrow-derived mesenchymal stem cells (BMSCs) under NSC-conditioned medium and neurotrophic factors. a, b, the generation of hESC-NSC according to published protocol. c, The immunostaining of Ki67 expression of BMSCs between regular L-DMEM complete medium, hESC-NSC conditioned medium. d, Bright-field images of expanded BMSCs after treated under regular L-DMEM complete medium, hESC-NSC conditioned medium, and Neurotrophic factors.

d) Together, these findings indicate that with the optimizations of neurotrophic supplements (NSs) for culture medium, distinct stem cell subpopulations could be induced and preserved for enhanced tissue regeneration.

Why devote so much effort (single cell analysis) to this specific formulation? The regenerative outcome was not even complete regeneration of the defect site, so how is this particular cocktail of more interest than comparably regenerative formulations?

Re: Thanks for the comment.

a) Both a complete coverage of the defect area and full-thickness reconstruction of the defect are required for the successful calvarial bone regeneration. Current approaches such as the implantation of biomineralized hydrogels [30] or osteogenic preconditioning[31] scaffolds with MSCs can hardly achieve a full-thickness reconstruction of the critical-sized calvarial bone defects, in which two layers of cortical bone plate and trabecular bone/bone marrow in between [32] exist, usually lead to the formation of a single thin-layer of immature bone tissue.

b) In our study, the micro-CT images are representative for each group and actually are at an average level. Still, a complete coverage of newly formed bone was also observed, even though not all samples are ideal like this, which is shown as below. Besides, a full-thickness regeneration of the calvarial/skull defects was achieved by NSs-loaded scaffold,

other than a single layer of disordered immature bone in vehicle scaffold or a thin layer of soft tissue in control Defect

μ -CT evaluation of bone regeneration in 3D printed scaffold containing NSs (Scaffold + NSs) groups at 3 months post-surgery (the yellow area indicates new bone formation)

c) Although lots of studies have adopted a local delivery of soluble factors for bone repair, the corresponding mechanisms during bone regenerative process at single-cell resolution are still not fully understood. Regarding the diversified types of neo-tissue formation and the structural restoration of the regenerated tissue in our results, we hypothesized that a unique atlas of in situ stem cells would be recruited or activated by this specific 3D printing tissue engineering graft. We then were encouraged to determine the differentially recruited and regulated cell subpopulations by scRNA-seq analysis under the transplantation of the NSs-loaded cell culture hydrogel system.

Also, at times optimized composition of the cocktail was indicated while this was not part of this study, nor optimized for in vivo application in cranial defects as used here.

Re: Thank you for the attentive review work.

The optimized composition of the cocktail used in this study was based on our preliminary results (data not shown). Our original intention of developing such an NSs-supplied medium was for high efficient MSC expansion in vitro. Since most of the ingredients are adapted from the medium supplements for MSC mesospheres [33] and neural stem cells [34] culture, we optimized the NSs cocktail by depleting some unnecessary recombinant growth factors that could induce cell differentiation, and then combined with fetal bovine serum (FBS) as our final formula for MSCs expansion and in vivo applications.

Finally, the composition of the NSs is disputable from the perspective that FBS is included which is known to have a large batch-to-batch variation that can have a major impact on any study in bone regeneration.

Re: Thanks for the professional comment. Indeed, the FBS is one of the most key elements for cell culture and the batch-to-batch variation exists theoretically. In our study, the NSs medium is relatively stable, which can be explained as follows:

a) Our previous results (data not shown) from FBS/serum-free version of NSs medium alone have shown a robust effect in promoting MSC proliferation, result is showed as below:

b) At least in 2 independent batches (GIBCO, Cat. NO .10099-141C, LOT. NO. 2206991CP & LOT.NO.2206992CP) of FBS was tested in our group, we didn't find any significant differences in cell expansion efficiency in vitro or huge variation of bone healing extent in animal experiments between batches.

The effect of FBS-free version of NSs medium for MSC culture

5. Description of results indicating significant differences are not always supported by actual quantitative data (Ki67, gammaH2AX, cell expansion/cell numbers, proportion of living cells) and therefore the associated conclusions regarding the local expansion of cellular subpopulations cannot be truly supported. Also, 2D quantification of chondrogenic differentiation is not very representative (3D cultures should be shown with preferably GAG but also collagen type II identification).

Re: Thanks for the suggestions.

a) We have provided the quantitative data of Ki67 and gammaH2AX to cell numbers, as shown in SFigure 1.

b) We have repeated the chondrogenic differentiation assay by 3D pellet culture model, and results from the SO Staining and immunostaining of collagen type II are highly indicative of the enhanced potential of chondrogenic differentiation of MSCs treated by NSs medium. The corresponding results were also revised in the manuscript in SFigure 1.

Assessment of the chondrogenic differentiation of MSC by 3D pellet culture model after NSs treatment. The upper panel shows the SO staining; and the lower panel shows immunostaining of collagen type II and SOX9 expression.

6. The cell expansion conclusion is not sufficiently supported by the data. FGF which is in the NSs for example, is known to increase cellular proliferation but also known to stimulate subsequent differentiation of cells. But only showing qualitative data on larger numbers of MSX1+ cells over time does not confirm targeted cell expansion of this subpopulation.

Re: We thank the reviewer for this comment.

a) We found that after treated with NSs, Msx1+ SSC (SSC2) significantly upregulated expression of Fgfr2, receptor of FGF, compared with other three skeletal stem/progenitor cells, indicating a specifically respond to NSs (figures are shown below).

Expression of Fgfr2 in four the skeletal stem/progenitor cells.

b) The results we observed in this study is based on defined time points, we cannot completely exclude the possibility of the expansion of other subpopulations at varied time points. Nevertheless, we indeed observed that Msx1+ SSC2 expanded specifically in 2 weeks. Also, our *in vitro* experiments showed as below:

NSs-loaded hydrogel culture system promotes the *in vitro* expansion of Msx1⁺ SSC2 subset. a, b. Violin plots showing Msx1 and Mmp13 expression levels in all cell subsets of regenerated skull tissue, respectively. **c.** Visualization of expression and co-expression of Msx1 and Mmp13 in UMAP plot. **d.** Experimental design for the *in vitro* expansion of freshly isolated rat bone marrow. **e.** Co-immunostaining of Msx1 and Mmp13 expression of primary bone marrow tissue after expansion for 2 weeks on the culture system of NSs-loaded 3D printed hydrogel scaffold (bar=30 μ m).

Was cell expansion not stimulated for all cells/subpopulations present or selectively for the MSX1+ population? And were the MSX1+ cells actually proliferating or acquiring the MSX1+ phenotype after proliferation. These data are not supplied and therefore the conclusions on cell expansion of a specific subtype not supported.

Re: Thanks for the comment. With regard to Msx1⁺ phenotype, we visualized expression level of Msx1 between NSs and control among eight subpopulations and results showed there hardly existed difference (as below). Yet, proportion of Msx1⁺ SSCs changed dramatically 2 weeks after NSs treatment. In other word, Msx1⁺ SSCs was selectively expanded after NSs treatment and may be correlated with time.

Expression of Msx1 in NSs and control cells among eight osteogenic cell populations.

7. Increased osteogenic capacity (ALP, ARS) were not normalized to cell numbers. Since cell expansion may be affected differently under the various conditions, this correction on the data is required.

Re: Thanks for the nice comment. Our previous results on the increased osteogenic capacity as indicated by ALP and ARS staining showed comparable level of positive staining when normalized to total cell numbers. We actually repeated the cell expansion and subsequent osteogenic differentiation assay. Similar to the previous results, our recent results on normalized ALP and ARS staining didn't show significant variation (results are shown below). In this case, we conclude that the advantage of NSs-supplemented medium for enhanced osteogenic capacity of MSC was largely due to the increased cell number.

Normalized ALP concentration

Normalized ALP concentration of expanded MSCs after osteogenic differentiation for 7 days.

Normalized ARS concentration

Normalized ARS value of expanded MSCs after osteogenic differentiation for 7 days.

8. Why were microchannels of 200um generated in the printed constructs? It is generally known that around 400um is optimal for bone regeneration purposes.

Re: Thanks for the comment.

a) In other study, interconnected porous microstructure of hydrogel (methacrylated glycol chitosan) with $150 \pm 50 \mu\text{m}$ showed significantly higher MSC cell viability when compared both with much smaller or bigger pore size. Additionally, scaffolds with a microchannel size around 200 μm provided more surface area for cell adhesion and spreading[9].

b) In our previous work, we found that the compressive strength of the 3D-printed hydrogel scaffold with a microchannel of 200 μm is much higher than the 300 μm and 400 μm groups[10].

9. I may have missed it but I would have liked to see a discussion of the presence of MSX1 on these cells. MSX1 is associated specifically with dental development and not typically associated with bone regeneration in the skeleton outside of the head region.

And if it's a chondro-repressor, how can it be co-expressed with col6 and aggrecan? Also, endochondral bone regeneration could be confirmed by staining for markers such as collagen type X.

Re: We thank the reviewer for this comment.

a) We have added a discussion of the presence of MSX1 on cells in the Discussion section labeled by yellow color.

b) After literature review, we are aware of that Msx1 was found played an early morphogenetic role during the calvarial bone development [35]. However, Msx1 expression extends into the postnatal stages of skull morphogenesis and is surprisingly involved in the endochondral bone formation in skeletal development [36]. Similarly, during the regenerative process of the calvarial bone injury, our single-cell data revealed that most of the Msx1⁺ SSCs are responsible for the osteogenic differentiation, which is illustrated in RNA velocity analysis (Fig 4). We found evidence of Acan and Col6 expression in our single-cell data, which is in the early stage (1 and 2 weeks after injury) of repair process (Fig 6). Moreover, in our lineage-tracing experiment, we also found a small proportion of Msx1⁺ SSCs got Col2 and Acan expression at four weeks after injury (Figure 7).

c) As endochondral bone regeneration could be confirmed by staining for markers such as collagen type X, we further stained the sections of regenerated tissue form rat calvarial defect model. Result showed the expression of Col10 was restricted in some area and partially overlapped with Msx1 expression.

Co-immunostaining of Msx1 and Col10 expression on paraffin sections in Untreated Defect group and Scaffold + NSs group at 2 weeks after defect surgery.

10. At some point a femur defect was introduced which was a bit confusing (Suppl fig 13). This was not described in the methods. Here local cells from the mesenchymal rather than neural crest origin will be recruited with a different differentiation capacity. Why were these data added and what are they showing? Were the recruited cells also MSX1+ here? What is the point of showing these data?

Re: Thanks for the reminding and suggestion. As data in the present study shows that Msx1+ SSCs subset may promote efficient bone regeneration partially through endochondral ossification, our original intention for the experiments for femur defect was to demonstrate the applicability in long bone defect model, where the skeletal stem cells actually orchestrate endochondral ossification for fracture repair. Since these extra results may cause confused understanding, we now have removed them from the current version of manuscript.

11. I would suggest rewriting of the manuscript by a native speaker since many errors are present, some of which hinder proper understanding of the manuscript.

Re: Thanks! We appreciated your patience and wise suggestions for reviewing this manuscript. We have now thoroughly rewritten and revised the manuscript, and the main figures, as well as improved the English language by the Springer Nature Author Services.

References:

1. Tomlinson, R. E., et al., *NGF-TrkA signaling in sensory nerves is required for skeletal adaptation to mechanical loads in mice*. Proc Natl Acad Sci U S A, 2017. **114**(18): p. E3632-E3641.
2. Meyers, C. A., et al., *A Neurotrophic Mechanism Directs Sensory Nerve Transit in Cranial Bone*. Cell Rep, 2020. **31**(8): p. 107696.
3. Nassif, A., et al., *Msx1 role in craniofacial bone morphogenesis*. Bone, 2014. **66**: p. 96-104.
4. Shah, P. T., et al., *Single-Cell Transcriptomics and Fate Mapping of Ependymal Cells Reveals an Absence of Neural Stem Cell Function*. Cell, 2018. **173**(4): p. 1045-1057 e9.
5. Qin, W., et al., *Osseointegration and biosafety of graphene oxide wrapped porous GF/PEEK composites as implantable materials: The role of surface structure and chemistry*. Dent Mater, 2020. **36**(10): p. 1289-1302.
6. Visser, J., et al., *Reinforcement of hydrogels using three-dimensionally printed microfibrils*. Nature Communications, 2015. **6**.
7. Hong, Y., et al., *A strongly adhesive hemostatic hydrogel for the repair of arterial and heart bleeds*. Nat Commun, 2019. **10**(1): p. 2060.
8. Hasani-Sadrabadi, M. M., et al., *An engineered cell-laden adhesive hydrogel promotes craniofacial bone tissue regeneration in rats*. Sci Transl Med, 2020. **12**(534).

9. Cui, Z.K., et al., *Microporous methacrylated glycol chitosan-montmorillonite nanocomposite hydrogel for bone tissue engineering*. Nat Commun, 2019. **10**(1): p. 3523.
10. Zhou, F., et al., *Rapid printing of bio-inspired 3D tissue constructs for skin regeneration*. Biomaterials, 2020. **258**: p. 120287.
11. Einhorn, T.A. and L.C. Gerstenfeld, *Fracture healing: mechanisms and interventions*. Nat Rev Rheumatol, 2015. **11**(1): p. 45–54.
12. Ji, C., et al., *Transcriptome Analysis Revealed the Symbiosis Niche of 3D Scaffolds to Accelerate Bone Defect Healing*. Adv Sci (Weinh), 2022. **9**(8): p. e2105194.
13. Herberg, S., et al., *Combinatorial morphogenetic and mechanical cues to mimic bone development for defect repair*. Sci Adv, 2019. **5**(8): p. eaax2476.
14. McDermott, A.M., et al., *Recapitulating bone development through engineered mesenchymal condensations and mechanical cues for tissue regeneration*. Science Translational Medicine, 2019. **11**(495).
15. Chan, C.K., et al., *Identification and specification of the mouse skeletal stem cell*. Cell, 2015. **160**(1–2): p. 285–98.
16. Debnath, S., et al., *Discovery of a periosteal stem cell mediating intramembranous bone formation*. Nature, 2018. **562**(7725): p. 133–139.
17. Tikhonova, A.N., et al., *The bone marrow microenvironment at single-cell resolution*. Nature, 2019. **569**(7755): p. 222–+.
18. Yamada, D., et al., *Induction and expansion of human PRRX1(+) limb-bud-like mesenchymal cells from pluripotent stem cells*. Nat Biomed Eng, 2021. **5**(8): p. 926–940.
19. Julien, A., et al., *Direct contribution of skeletal muscle mesenchymal progenitors to bone repair*. Nat Commun, 2021. **12**(1): p. 2860.
20. Li, Z., et al., *Fracture repair requires TrkA signaling by skeletal sensory nerves*. J Clin Invest, 2019. **129**(12): p. 5137–5150.
21. Tomlinson, R.E., et al., *NGF-TrkA Signaling by Sensory Nerves Coordinates the Vascularization and Ossification of Developing Endochondral Bone*. Cell Rep, 2016. **16**(10): p. 2723–2735.
22. Thrivikraman, G., et al., *Rapid fabrication of vascularized and innervated cell-laden bone models with biomimetic intrafibrillar collagen mineralization*. Nat Commun, 2019. **10**(1): p. 3520.
23. Chai, Y. and R.E. Maxson, Jr., *Recent advances in craniofacial morphogenesis*. Dev Dyn, 2006. **235**(9): p. 2353–75.
24. Maruyama, T., et al., *Stem cells of the suture mesenchyme in craniofacial bone development, repair and regeneration*. Nat Commun, 2016. **7**: p. 10526.
25. Tomlinson, R.E., et al., *NGF-TrkA signaling in sensory nerves is required for skeletal adaptation to mechanical loads in mice*. Proceedings of the National Academy of Sciences of the United States of America, 2017. **114**(18): p. E3632–E3641.
26. Li, Z., et al., *Fracture repair requires TrkA signaling by skeletal sensory nerves*. Journal of Clinical Investigation, 2019. **129**(12): p. 5137–5150.

27. Meyers, C.A., et al., *A Neurotrophic Mechanism Directs Sensory Nerve Transit in Cranial Bone*. Cell Reports, 2020. **31**(8).
28. Isern, J., et al., *Self-Renewing Human Bone Marrow Mesospheres Promote Hematopoietic Stem Cell Expansion*. Cell Reports, 2013. **3**(5): p. 1714-1724.
29. Tietze, S., et al., *Spheroid Culture of Mesenchymal Stromal Cells Results in Morphorheological Properties Appropriate for Improved Microcirculation (vol 6, 1802104, 2019)*. Advanced Science, 2019. **6**(8).
30. Cui, Z.K., et al., *Microporous methacrylated glycol chitosan-montmorillonite nanocomposite hydrogel for bone tissue engineering*. Nature Communications, 2019. **10**.
31. Harvestine, J.N., et al., *Osteogenic preconditioning in perfusion bioreactors improves vascularization and bone formation by human bone marrow aspirates*. Science Advances, 2020. **6**(7).
32. Cugurra, A., et al., *Skull and vertebral bone marrow are myeloid cell reservoirs for the meninges and CNS parenchyma*. Science, 2021. **373**(6553): p. 409-+.
33. Mendez-Ferrer, S., et al., *Mesenchymal and haematopoietic stem cells form a unique bone marrow niche*. Nature, 2010. **466**(7308): p. 829-U59.
34. Kumamaru, H., et al., *Generation and post-injury integration of human spinal cord neural stem cells*. Nature Methods, 2018. **15**(9): p. 723-+.
35. Dash, S. and P.A. Trainor, *The development, patterning and evolution of neural crest cell differentiation into cartilage and bone*. Bone, 2020. **137**.
36. Orestes-Cardoso, S.M., et al., *Postnatal Msx1 expression pattern in craniofacial, axial, and appendicular skeleton of transgenic mice from the first week until the second year*. Dev Dyn, 2001. **221**(1): p. 1-13.

REVIEWERS' COMMENTS

Reviewer #1 (Remarks to the Author):

The authors have now answered adequately to the reviewers comments and issues raised and the quality of the manuscript improved significantly.

Reviewer #2 (Remarks to the Author):

The authors have comprehensively revised the manuscript and addressed all of my queries. I have one more comment & question about their characterization of the populations labeled SSC1 and SSC2. I have had difficulty interpreting their characterization outlined in their response (item (b) on pages 6 and 7). Both studies cited here (by Longaker and Greenblatt labs) indicate lack of CD105 expression as a requirement for a cell population to be defined as stem cells. Yet, both SSC populations identified by the authors express CD105. Further, emerging data by the Kalajzic Lab indicate that the skeletal stem cell definition proposed by the Longaker Lab does not exclude mature osteoblasts, suggesting that this definition requires further refinement. All this being said, in the absence of direct functional data (which would demonstrate a capacity for self-renewal and multi-lineage potential), I think it is inappropriate to use the term "stem cell" for populations SSC1 and SSC2. I recommend that the authors acknowledge these shortcomings in their annotation attempts, and define these cells otherwise, perhaps as progenitors to be further characterized in the future.

Other minor comments:

Line 129 – It would be nice to have a brief description of your optimized formulation

Line 204 – Please indicate the animal model you are using here.

Line 235 – Have you quantified these changes?

Reviewer #3 (Remarks to the Author):

Reviewer 3:

The authors have carefully addressed my comments. A few points are not fully clear yet:

point 1: Yes, calvarial bone plates are FORMED via intramembranous ossification. What I tried to point out here is that REPAIR or REGENERATION of a defect can still take place via intramembranous AND/OR endochondral mechanisms. Please adapt accordingly in your manuscript.

point 2: some unclarities remain:

a. page 4, line 95 – explain mesosphere; line 96 – what are improved microcirculation properties and are those tested in vivo?

b. Unclear why neurotrophic factors are chosen while these are known for stimulating neurons – what is the rationale to not use a cocktail of factors that are known to stimulate bone formation (TGFs and BMPs for example)

c. You refer to skeletal stem cell subpopulations regularly and also osteogenic subpopulations – which subpopulations (markers) do you mean exactly?

d. 3D-printing is coming out of the blue in your hypothesis – why do you need to print your samples?

Point 4:

4a. If the NSs were previously mainly shown to support innervation, what was your hypothesis to use this for bone regeneration? Please include a rationale in your introduction. As you know, NSs are not essential for osteoprogenitor expansion, so why go for a neuro-oriented cocktail and not for more commonly used osteo-stimulatory agents (BMPs etc)?

4b. Enhancing MSC expansion with such growth factors is not novel in my opinion. In general, your description of the novelty is insufficient in the manuscript. For example, the new yellow first sentence of the discussion is not indicating a novel finding of your study. This comment is based on your previous work and not specific for the results of this particular study.

Point 11: further revision may be required to ease the reading of the manuscript and to check the newly inserted yellow texts (they also contain errors).

Point-by-point response to the reviewers' comments

REVIEWERS' COMMENTS

Reviewer #1 (Remarks to the Author):

The authors have now answered adequately to the reviewers comments and issues raised and the quality of the manuscript improved significantly.

Re: Thanks for your patient review work.

Reviewer #2 (Remarks to the Author):

The authors have comprehensively revised the manuscript and addressed all of my queries. I have one more comment & question about their characterization of the populations labeled SSC1 and SSC2. I have had difficulty interpreting their characterization outlined in their response (item (b) on pages 6 and 7). Both studies cited here (by Longaker and Greenblatt labs) indicate lack of CD105 expression as a requirement for a cell population to be defined as stem cells. Yet, both SSC populations identified by the authors express CD105.

Further, emerging data by the Kalajzic Lab indicate that the skeletal stem cell definition proposed by the Longaker Lab does not exclude mature osteoblasts, suggesting that this definition requires further refinement. All this being said, in the absence of direct functional data (which would demonstrate a capacity for self-renewal and multi-lineage potential), I think it is inappropriate to use the term "stem cell" for populations SSC1 and SSC2. I recommend that the authors acknowledge these shortcomings in their annotation attempts, and define these cells otherwise, perhaps as progenitors to be further characterized in the future.

Re: Thank you again for your professional comments and advice. The SSC subpopulations that defined in this study do not include mature osteoblasts from our scRNA-Seq data analysis, as shown in SFigure 11, where a separate osteoblast subset was identified. Yes, we accept that there are shortcomings in the annotations for SSCs, which need to be further characterized in the future work. Accordingly, we have acknowledged the shortcomings in the annotations for the skeletal stem/progenitor cells in the third paragraph of the Discussion section.

Other minor comments:

Line 129 – It would be nice to have a brief description of your optimized formulation

Re: Thanks for your nice reminding. Actually we have provided the optimized formulation as "Neurotrophic Supplements (NSs)" in the "Methods" section of the revised manuscript. To avoid further possible confuse, we have also inserted the "NSs" in the Line 129 shown as "our final NSs formula for MSC expansion", which is labelled for yellow color.

Line 204 – Please indicate the animal model you are using here.

Re: Thanks. The animal model used here is subcutaneous implantation of our NSs-loaded in situ culture system at the dorsal site of SD rat species. The detailed description was also shown in the second paragraph of “Animal and surgical procedures” of the “Methods” section.

Line 235 – Have you quantified these changes?

Re: Thanks for your comments and brilliant advice. We have provided the relatively quantitative data of OCN-positive matrix immunostaining in the Supplementary Fig. 4c, d of the Revised Supplementary Information.

Reviewer #3 (Remarks to the Author):

Reviewer 3:

The authors have carefully addressed my comments. A few points are not fully clear yet: point 1: Yes, calvarial bone plates are FORMED via intramembranous ossification. What I tried to point out here is that REPAIR or REGENERATION of a defect can still take place via intramembranous AND/OR endochondral mechanisms. Please adapt accordingly in your manuscript.

Re: Thank you again for your professional advice. To this point, we are fully agreed with this opinion. We have revised the manuscript accordingly as required; please refer to the deleted description in “Abstract” and the last paragraph in the “Introduction”.

point 2: some unclarities remain:

a. page 4, line 95 – explain mesosphere; line 96 – what are improved microcirculation properties and are those tested in vivo?

Re: Thank you again for your comments.

1) Mesosphere: Fetal human bone marrow mononuclear cells were immunomagnetically depleted of CD45+ cells and plated in mesosphere medium. Numerous human primary spheres formed after 7–10 days in culture, the mesospheres were capable of robustly differentiating into osteoblasts, adipocytes, and chondrocytes[1].
2) Microcirculation properties that were referred in the work[2] including multipotent, self-Renewable, and immunomodulatory properties, especially cellular morphorheological properties, are improved after the specific culture with the medium. These altered properties were subsequently tested in vivo by the intravenous injection. Results showed that, at multi-organ level, the mesosphere MSCs can pass lung capillaries more efficiently resulting in a more even distribution within the capillary networks of other tissues. Compared to 2D cultured MSCs, lung trapping of mesosphere MSCs decreased by about 30%, whereas recovery in liver, heart, spleen, and kidney increased up to 20%

b. Unclear why neurotrophic factors are chosen while these are known for stimulating neurons – what is the rationale to not use a cocktail of factors that are known to stimulate bone formation (TGFs and BMPs for example)

Re: Thank you again for your comments. Cocktail of factors including TGFs and BMPs are known to stimulate bone formation by promoting the osteogenic differentiation of the mesenchymal stem cells. However, the therapeutic effect of these bioactive factors was largely affected by the harmful ectopic bone formation, osteoclast activation and soft tissue inflammation[3], as well as the requirements for additional mechanical stimuli. While the in vivo stem/progenitor cells during bone regeneration process are also highly heterogeneous, biomedical engineering strategies by targeting in situ expansion of specific stem cell subpopulations with osteogenic potential remains challenging. In our previous results, we found that the proliferation of bone marrow mesenchymal stem cells (BMSCs) was significantly increased by using neural stem cells (NSCs) conditioned medium compared to the regular L-DMEM complete medium. Subsequently, we found a more stronger BMSCs proliferation when directly adding the cocktail of neurotrophic factors to the culture medium. Thus, we hypothesize that in situ delivery of NSs through a 3D-printed bioactive hydrogel graft could induce bone regeneration by promoting local expansion of skeletal stem cell subpopulations in vivo.

c. You refer to skeletal stem cell subpopulations regularly and also osteogenic subpopulations – which subpopulations (markers) do you mean exactly?

Re: Thank you again for your comments. The exact subpopulation we are focusing is the “Skeletal stem cell 2” with Msx1 and Mmp13 double positive, and the markers are which is mainly shown in the Figure 4 and 5.

d. 3D-printing is coming out of the blue in your hypothesis – why do you need to print your samples?

Re: Thank you again for your comments. In our previous studies, this GelMA/HA-NB/LAP hydrogel was used as bio-adhesive patch for repair of large skull defects, results showed that pure hydrogel without porous structure cannot fully regenerated the bone defect, which is due to the limited stem cell migration [An elastic auto-bone patch for one-step repair large skull defects accompanied by Craniocerebral injury]. And our more recent work has shown that by 3D-printing technology the printed scaffold holds microporous structures, and is beneficial for neovascularization and endogenous cell migration [4], which are required for efficient bone formation.

For the critical-sized bone defects, a hydrogel scaffold with defined porous structure and mechanical strength was preferred than direct injection or implantation, to provide more adhering surfaces and higher mechanical support for endogenous stem cells and blood vessel cells growth and migration during the bone repair process [5]. Besides, a form of 3D-printed hydrogel scaffold enables more efficient oxygen and nutrient exchange, and is beneficial for the necessary vascularization, osteogenic differentiation, and the subsequent calcium deposition [6] during bone regeneration.

Point 4:

4a. If the NSs were previously mainly shown to support innervation, what was your

hypothesis to use this for bone regeneration? Please include a rationale in your introduction. As you know, NSs are not essential for osteoprogenitor expansion, so why go for a neuro-oriented cocktail and not for more commonly used osteo-stimulatory agents (BMPs etc)?

Re: Thank you again for your comments. Neurotrophic factors, for example NGFs, were previously shown to support innervation, not for osteoprogenitor expansion. In this study, our hypothesis is that a cocktail of neurotrophic supplements, including a combination of multiple growth factors, promotes efficient bone regeneration through direct expansion of distinct stem cell subsets. We have included this rationale in the introduction section, which is labelled by yellow color.

While osteo-stimulatory agents, for example BMPs, were largely beneficial for promoting the osteogenic differentiation of endogenous stem/progenitor cells. However, the therapeutic efficacy is often limited under the condition of critical-sized bone defects. Besides, the therapeutic effect of BMPs delivery is largely limited by harmful ectopic bone formation, osteoclast activation and soft tissue inflammation.

4b. Enhancing MSC expansion with such growth factors is not novel in my opinion. In general, your description of the novelty is insufficient in the manuscript. For example, the new yellow first sentence of the discussion is not indicating a novel finding of your study. This comment is based on your previous work and not specific for the results of this particular study.

Re: Thank you again for your wise suggestions. We are agreed with the opinion that our novelty should be based on our current work. And we have revised the statements in the discussion section that labelled by yellow color.

Point 11: further revision may be required to ease the reading of the manuscript and to check the newly inserted yellow texts (they also contain errors).

Re: Thank you again for your comments. we have revised the manuscript again accordingly.

References:

1. Isern, J., et al., *Self-renewing human bone marrow mesospheres promote hematopoietic stem cell expansion*. Cell Rep, 2013. **3**(5): p. 1714-24.
2. Tietze, S., et al., *Spheroid Culture of Mesenchymal Stromal Cells Results in Morphological Properties Appropriate for Improved Microcirculation*. Adv Sci (Weinh), 2019. **6**(8): p. 1802104.
3. Boerckel, J.D., et al., *Effects of protein dose and delivery system on BMP-mediated bone regeneration*. Biomaterials, 2011. **32**(22): p. 5241-5251.
4. Zhou, F., et al., *Rapid printing of bio-inspired 3D tissue constructs for skin regeneration*. Biomaterials, 2020. **258**: p. 120287.
5. Qin, W., et al., *Osseointegration and biosafety of graphene oxide wrapped porous CF/PEEK composites as implantable materials: The role of surface structure and chemistry*. Dent Mater, 2020. **36**(10): p. 1289-1302.

6. Visser, J., et al., *Reinforcement of hydrogels using three-dimensionally printed microfibrils*. Nature Communications, 2015. **6**.